# The Mouse Action Recognition System (MARS) software pipeline for automated analysis of social behaviors in mice

Cristina Segalin[1], Jalani Williams[1], Tomomi Karigo[2], May Hui[2], Moriel Zelikowsky[2†], Jennifer J Sun[1], Pietro Perona[1], David J Anderson[2,3], Ann Kennedy[2*‡]

[1]Department of Computing & Mathematical Sciences, California Institute of Technology, Pasadena, United States; [2]Division of Biology and Biological Engineering 156-29, TianQiao and Chrissy Chen Institute for Neuroscience, California Institute of Technology, Pasadena, United States; [3]Howard Hughes Medical Institute, California Institute of Technology, Pasadena, United States

**Abstract** The study of naturalistic social behavior requires quantification of animals' interactions. This is generally done through manual annotation—a highly time-consuming and tedious process. Recent advances in computer vision enable tracking the pose (posture) of freely behaving animals. However, automatically and accurately classifying complex social behaviors remains technically challenging. We introduce the Mouse Action Recognition System (MARS), an automated pipeline for pose estimation and behavior quantification in pairs of freely interacting mice. We compare MARS's annotations to human annotations and find that MARS's pose estimation and behavior classification achieve human-level performance. We also release the pose and annotation datasets used to train MARS to serve as community benchmarks and resources. Finally, we introduce the Behavior Ensemble and Neural Trajectory Observatory (BENTO), a graphical user interface for analysis of multimodal neuroscience datasets. Together, MARS and BENTO provide an end-to-end pipeline for behavior data extraction and analysis in a package that is user-friendly and easily modifiable.

**\*For correspondence:**
ann.kennedy@northwestern.edu

**Present address:** †Department of Neurobiology and Anatomy, University of Utah, Salt Lake City, United States; ‡Department of Neuroscience, Northwestern University Feinberg School of Medicine, Chicago, United States

**Competing interest:** The authors declare that no competing interests exist.

## Introduction

The brain evolved to guide survival-related behaviors, which frequently involve interaction with other animals. Gaining insight into brain systems that control these behaviors requires recording and manipulating neural activity while measuring behavior in freely moving animals. Recent technological advances, such as miniaturized imaging and electrophysiological devices, have enabled the recording of neural activity in freely behaving mice (*Remedios et al., 2017*; *Li et al., 2017*; *Falkner et al., 2020*)—however, to make sense of the recorded neural activity, it is also necessary to obtain a detailed characterization of the animals' actions during recording. This is usually accomplished via manual scoring of the animals' actions (*Yang et al., 2011*; *Silverman et al., 2010*; *Winslow, 2003*). A typical study of freely behaving animals can produce tens to hundreds of hours of video that require manual behavioral annotation (*Zelikowsky et al., 2018*; *Shemesh et al., 2013*; *Branson et al., 2009*). Scoring for social behaviors often takes human annotators 3–4× the video's duration to annotate; for long recordings, there is also risk of drops in annotation quality due to drifting annotator attention. It is unclear to what extent individual human annotators within and between different labs agree on the definitions of behaviors, especially the precise timing of behavior onset/offset. When behavior is being analyzed alongside neural recording data, it is also often unclear whether the set of social behaviors that were chosen to annotate are a good fit for explaining the activity of a neural population or whether other, unannotated behaviors with clearer neural correlates may have been missed.

An accurate, sharable, automated approach to scoring social behavior is thus needed. Use of such a pipeline would enable social behavior measurements in large-scale experiments (e.g., genetic or drug screens), and comparison of datasets generated across the neuroscience community by using a common set of definitions and classification methods for behaviors of interest. Automation of behavior classification using machine learning methods poses a potential solution to both the time demand of annotation and to the risk of inter-individual and inter-lab differences in annotation style.

We present the Mouse Action Recognition System (MARS), a quartet of software tools for automated behavior analysis, training and evaluation of novel pose estimator and behavior classification models, and joint visualization of neural and behavioral data (*Figure 1*). This software is accompanied by three datasets aimed at characterizing inter-annotator variability for both pose and behavior annotation. Together, the software and datasets introduced in this paper provide a robust computational pipeline for the analysis of social behavior in pairs of interacting mice and establish essential measures of reliability and sources of variability in human annotations of animal pose and behavior.

## Contributions

The contributions of this paper are as follows:

### Data

MARS pose estimators are trained on a novel corpus of manual pose annotations in top- and front-view video (*Figure 1—figure supplement 1*) of pairs of mice engaged in a standard resident-intruder assay (*Thurmond, 1975*). These data include a variety of experimental manipulations of the resident animal, including mice that are unoperated, cannulated, or implanted with fiberoptic cables, fiber photometry cables, or a head-mounted microendoscope, with one or more cables leading from the animal's head to a commutator feeding out the top of the cage. All MARS training datasets can be found at https://neuroethology.github.io/MARS/ under 'datasets.'

### Multi-annotator pose dataset

Anatomical landmarks ('keypoints' in the following) in this training set are manually annotated by five human annotators, whose labels are combined to create a 'consensus' keypoint location for each image. 9 anatomical keypoints are annotated on each mouse in the top view, and 13 in the front view (two keypoints, corresponding to the midpoint and end of the tail, are included in this dataset but were omitted in training MARS due to high annotator noise).

### Behavior classifier training/testing dataset

MARS includes three supervised classifiers trained to detect attack, mounting, and close investigation behaviors in tracked animals. These classifiers were trained on 6.95 hr of behavior video, 4 hr of which were obtained from animals with a cable-attached device such as a microendoscope. Separate evaluation (3.85 hr) and test (3.37 hr) sets of videos were used to constrain training and evaluate MARS performance, giving a total of over 14 hr of video (*Figure 1—figure supplement 2*). All videos were manually annotated on a frame-by-frame basis by a single trained human annotator. Most videos in this dataset are a subset of the recent CalMS mouse social behavior dataset (*Sun et al., 2021a*) (specifically, from Task 1).

### Multi-annotator behavior dataset

To evaluate inter-annotator variability in behavior classification, we also collected frame-by-frame manual labels of animal actions by eight trained human annotators on a dataset of ten 10-min videos. Two of these videos were annotated by all eight annotators a second time a minimum of 10 months later for evaluation of annotator self-consistency.

### Software

This paper is accompanied by four software tools, all of which can be found on the MARS project website at: https://neuroethology.github.io/MARS/.

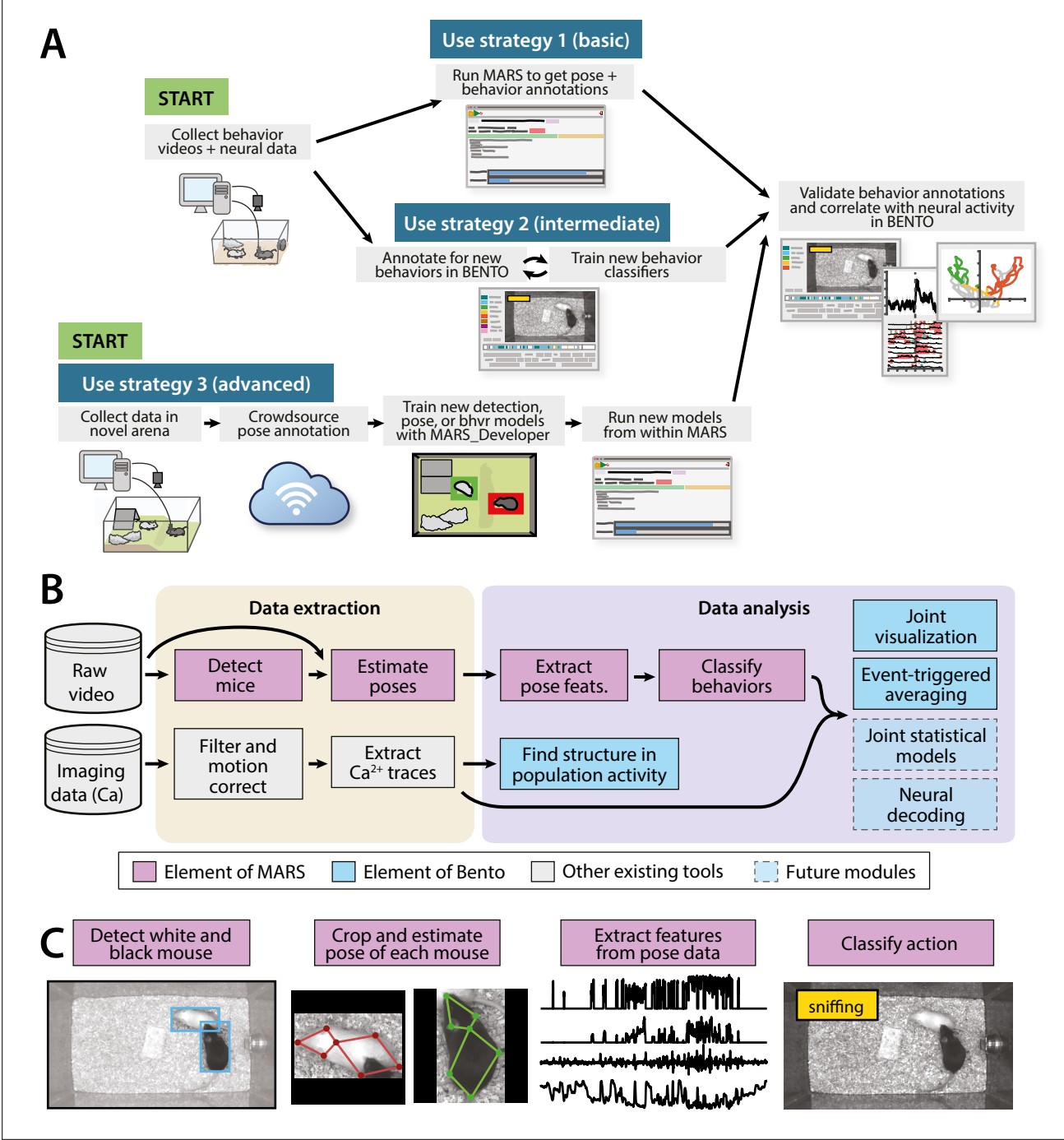

**Figure 1.** The Mouse Action Recognition System (MARS) data pipeline. (**A**) Sample use strategies of MARS, including either out-of-the-box application or fine-tuning to custom arenas or behaviors of interest. (**B**) Overview of data extraction and analysis steps in a typical neuroscience experiment, indicating contributions to this process by MARS and Behavior Ensemble and Neural Trajectory Observatory (BENTO). (**C**) Illustration of the four stages of data processing included in MARS.

The online version of this article includes the following figure supplement(s) for figure 1:

**Figure supplement 1.** Mouse Action Recognition System (MARS) camera positioning and sample frames.

**Figure supplement 2.** The Mouse Action Recognition System (MARS) annotation dataset.

**Figure supplement 3.** Mouse Action Recognition System (MARS) graphical user interface.

## MARS

An open-source, Python-based tool for running trained detection, pose estimation, and behavior classification models on video data. MARS can be run on a desktop computer equipped with TensorFlow and a graphical processing unit (GPU), and supports both Python command-line and graphical user interface (GUI)-based usage (*Figure 1—figure supplement 3*). The MARS GUI allows users to select a directory containing videos and will produce as output a folder containing bounding boxes, pose estimates, features, and predicted behaviors for each video in the directory.

## MARS_Developer

Python suite for training MARS on new datasets and behaviors. It includes the following components: (1) a module for collecting crowdsourced pose annotation datasets, (2) a module for training a MultiBox detector, (3) a module for training a stacked hourglass network for pose estimation, and (4) a module for training new behavior classifiers. It is accompanied by a Jupyter notebook guiding users through the training process.

## MARS_pycocotools

A fork of the popular COCO API for evaluation of object detection and pose estimation models (*Lin et al., 2014*), used within MARS_Developer. In addition to the original COCO API, it includes added scripts for quantifying performance of keypoint-based pose estimates, as well as added support for computing object keypoint similarity (OKS) scores (see Materials and methods) in laboratory mice.

## The Behavior Ensemble and Neural Trajectory Observatory (BENTO)

A MATLAB-based GUI for synchronous display of neural recording data, multiple videos, human/automated behavior annotations, spectrograms of recorded audio, pose estimates, and 270 'features' extracted from MARS pose data—such as animals' velocities, joint angles, and relative positions. It features an interface for fast frame-by-frame manual annotation of animal behavior, as well as a tool to create annotations programmatically by applying thresholds to combinations of the MARS pose features. BENTO also provides tools for exploratory neural data analysis, such as PCA and event-triggered averaging. While BENTO can be linked to MARS to annotate and train classifiers for behaviors of interest, BENTO may also be used independently, and with plug-in support can be used to display pose estimates from other systems such as DeepLabCut (*Mathis et al., 2018*).

## Related work

Automated tracking and behavior classification can be broken into a series of computational steps, which may be implemented separately, as we do, or combined into a single module. First, animals are detected, producing a 2D/3D centroid, blob, or bounding box that captures the animal's location, and possibly its orientation. When animals are filmed in an empty arena, a common approach is to use background subtraction to segment animals from their environments (*Branson et al., 2009*). Deep networks for object detection (such as Inception Resnet [*Szegedy et al., 2017*], Yolo [*Redmon et al., 2016*], or Mask R-CNN [*He et al., 2017*]) may also be used. Some behavior systems, such as Ethovision (*Noldus et al., 2001*), MoTr (*Ohayon et al., 2013*), idTracker (*Pérez-Escudero et al., 2014*), and previous work from our group (*Hong et al., 2015*), classify behavior from this location and movement information alone. MARS uses the MSC-MultiBox approach to detect each mouse prior to pose estimation; this architecture was chosen for its combined speed and accuracy.

The tracking of multiple animals raises problems not encountered in single-animal tracking systems. First, each animal must be detected, located, and identified consistently over the duration of the video. Altering the appearance of individuals using paint or dye, or selecting animals with differing coat colors, facilitates this task (*Ohayon et al., 2013*, *Gal et al., 2020*). In cases where these manipulations are not possible, animal identity can in some cases be tracked by identity-matching algorithms (*Branson et al., 2009*). The pretrained version of MARS requires using animals of differing coat colors (black and white).

Second, the posture ('pose') of the animal, including its orientation and body part configuration, is computed for each frame and tracked across frames. A pose estimate comprises the position and identity of multiple tracked body parts, either in terms of a set of anatomical 'keypoints' (*Toshev and Szegedy, 2014*), shapes (*Dankert et al., 2009*, *Dollár et al., 2010*), or a dense 2D or 3D mesh (*Güler*

*et al., 2018*). Keypoints are typically defined based on anatomical landmarks (nose, ears, paws, digits), and their selection is determined by the experimenter depending on the recording setup and type of motion being tracked.

Animal tracking and pose estimation systems have evolved in step with the field of computer vision. Early computer vision systems relied on specialized data acquisition setups using multiple cameras and/or depth sensors (*Hong et al., 2015*), and were sensitive to minor changes in experimental conditions. More recently, systems for pose estimation based on machine learning and deep neural networks, including DeepLabCut (*Mathis et al., 2018*), LEAP (*Pereira et al., 2019*), and Deep-PoseKit (*Graving et al., 2019*), have emerged as a flexible and accurate tool in behavioral and systems neuroscience. These networks, like MARS's pose estimator, are more accurate and more adaptable to recording changes than their predecessors (*Sturman et al., 2020*), although they require an initial investment in creating labeled training data before they can be used.

Third, once raw animal pose data are acquired, a classification or identification of behavior is required. Several methods have been introduced for analyzing the actions of animals in an unsupervised or semi-supervised manner, in which behaviors are identified by extracting features from the animal's pose and performing clustering or temporal segmentation based on those features, including Moseq (*Wiltschko et al., 2015*), MotionMapper (*Berman et al., 2014*), and multiscale unsupervised structure learning (*Vogelstein et al., 2014*). Unsupervised techniques are said to identify behaviors in a 'user-unbiased' manner (although the behaviors identified do depend on how pose is preprocessed prior to clustering). Thus far, they are most successful when studying individual animals in isolation.

Our goal is to detect complex and temporally structured social behaviors that were previously determined to be of interest to experimenters; therefore, MARS takes a supervised learning approach to behavior detection. Recent examples of supervised approaches to detection of social behavior include *Giancardo et al., 2013*, MiceProfiler (*de Chaumont et al., 2012*), SimBA (*Nilsson, 2020*), and *Hong et al., 2015*. Like MARS, SimBA uses a keypoint-based representation of animal pose, obtained via separate software (supported pose representations include DeepLabCut [*Mathis et al., 2018*], DeepPoseKit [*Graving et al., 2019*], SLEAP [*Pereira et al., 2020b*], and MARS itself). In contrast, Giancardo et al., Hong et al., and MiceProfiler are pre-deep-learning methods that characterize animal pose in terms of geometrical primitives (*Hong et al., 2015*, *Giancardo et al., 2013*) or contours extracted using background subtraction (*de Chaumont et al., 2012*). Following pose estimation, all five systems extract a set of handcrafted spatiotemporal features from animal pose: features common to all systems include relative position, animal shape (typically body area), animal movement, and inter-animal orientation. MARS and Hong et al. use additional handcrafted features capturing the orientation and minimum distances between interacting animals. Both MARS and SimBA adopt the rolling feature-windowing method introduced by JAABA (*Kabra et al., 2013*), although choice of windowing differs modestly: SimBA computes raw and normalized feature median, mean, and sum within five rolling time windows, whereas MARS computes feature mean, standard deviation, minimum, and maximum values, and uses three windows. Finally, most methods use these handcrafted features as inputs to trained ensemble-based classifiers: Adaptive Boosting in Hong et al., Random Forests in SimBA, Temporal Random Forests in Giancardo et al., and Gradient Boosting in MARS; MiceProfiler instead identifies behaviors using handcrafted functions. While there are many similarities between the approaches of these tools, direct comparison of performance is challenging due to lack of standardized evaluation metrics. We have attempted to address this issue in a separate paper (*Sun et al., 2021a*).

A last difference between these five supervised approaches is their user interface and flexibility. Three are designed for out-of-the-box use in single, fixed settings: Giancardo et al. and Hong et al. in the resident-intruder assay, and MiceProfiler in a large open-field arena. SimBA is fully user-defined, functioning in diverse experimental arenas but requiring users to train their own pose estimation and behavior models; a GUI is provided for this purpose. MARS takes a hybrid approach: whereas the core 'end-user' version of MARS provides pretrained pose and behavior models that function in a standard resident-intruder assay, MARS_Developer allows users to train MARS pose and behavior models for their own applications. Unique to MARS_Developer is a novel library for collecting crowdsourced pose annotation datasets, including tools for quantifying inter-human variability in pose labels and using this variability to evaluate trained pose models. The BENTO GUI accompanying MARS is also unique: while BENTO does support behavior annotation and (like SimBA) behavior classifier training,

it is aimed primarily at exploratory analysis of multimodal datasets. In addition to pose and annotation data, BENTO can display neural recordings and audio spectrograms, and supports basic neural data analyses such as event-triggered averaging, k-means clustering, and dimensionality reduction.

Lastly, supervised behavior classification can also be performed directly from video frames, forgoing the animal detection and pose estimation steps (*Monfort et al., 2020*, *Tran et al., 2015*). This is usually done by adopting variations of convolutional neural networks (CNNs) to classify frame-by-frame actions or combining CNN and recurrent neural network (RNN) architectures that classify the full video as an action or behavior, and typically requires many more labeled examples than pose-based behavior classification. We chose a pose-based approach for MARS both because it requires fewer training examples and because we find that the intermediate step of pose estimation is useful in its own right for analyzing finer features of animal behavior and is more interpretable than features extracted by CNNs directly from video frames.

## Results

### Characterizing variability of human pose annotations

We investigated the degree of variability in human annotations of animal pose for two camera placements—filming animal interactions from above and from the front—using our previously published recording chamber (*Hong et al., 2015*; *Figure 1—figure supplement 1A*). We collected 93 pairs of top- and front-view behavior videos (over 1.5 million frames per view) under a variety of lighting/camera settings, bedding conditions, and experimental manipulations of the recorded animals (*Figure 1—figure supplement 1B*). A subset of 15,000 frames were uniformly sampled from each of the top- and front-view datasets, and manually labeled by trained human annotators for a set of anatomically defined keypoints on the bodies of each mouse (*Figure 2A and B*, see Materials and methods for description of annotator workforce). 5000 frames in our labeled dataset are from experiments in which the black mouse was implanted with either a microendoscopic, fiber photometry, or optogenetic system attached to a cable of varying color and thickness. This focus on manipulated mice allowed us to train pose estimators to be robust to the presence of devices or cables.

To assess annotator reliability, each keypoint in each frame was annotated by five individuals; the median across annotations was taken to be the ground-truth location of that keypoint as we found this approach to be robust to outliers (*Figure 2C–F*, see Materials and methods). To quantify annotator variability, we adapted the widely used percent correct Keypoints (PCK) metric used for pose estimate evaluation (*Yang and Ramanan, 2013*). First, for each frame, we computed the distance of each keypoint by each annotator to the median keypoint location across the remaining four annotators. Next, we computed the percentage of frames for which the annotator closest to ground truth on a given frame was within a radius X, over a range of values of X (*Figure 2G and H*, blue lines). Finally, we repeated this calculation using the annotator furthest from ground truth on each frame (green lines) and the average annotator distance to ground truth on each frame (orange lines), thus giving a sense of the range of human performance in pose annotation. We observed much higher inter-annotator variability for front-view videos compared to top-view videos: 86.2% of human-annotated keypoints fell within a 5 mm radius of ground truth in top-view frames, while only 52.3% fell within a 5 mm radius of ground truth in front-view frames (scale bar in *Figure 2E and F*). Higher inter-annotator variability in the front view likely arises from the much higher incidence of occlusion in this view, as can be seen in the sample frames in *Figure 1—figure supplement 1B*.

### Pose estimation of unoperated and device-implanted mice in the resident-intruder assay

We used our human-labeled pose dataset to train a machine learning system for pose estimation in interacting mice. While multiple pose estimation systems exist for laboratory mice (*Mathis et al., 2018*; *Pereira et al., 2019*; *Graving et al., 2019*), we chose to include a novel pose estimation system within MARS for three reasons: (1) to produce an adequately detailed representation of the animal's posture, (2) to allow integration of MARS with existing tools that detect mice (in the form of bounding boxes) but do not produce detailed pose estimates, and (3) to ensure high-quality pose estimation in cases of occlusion and motion blur during social interactions. Pose estimation in MARS is carried

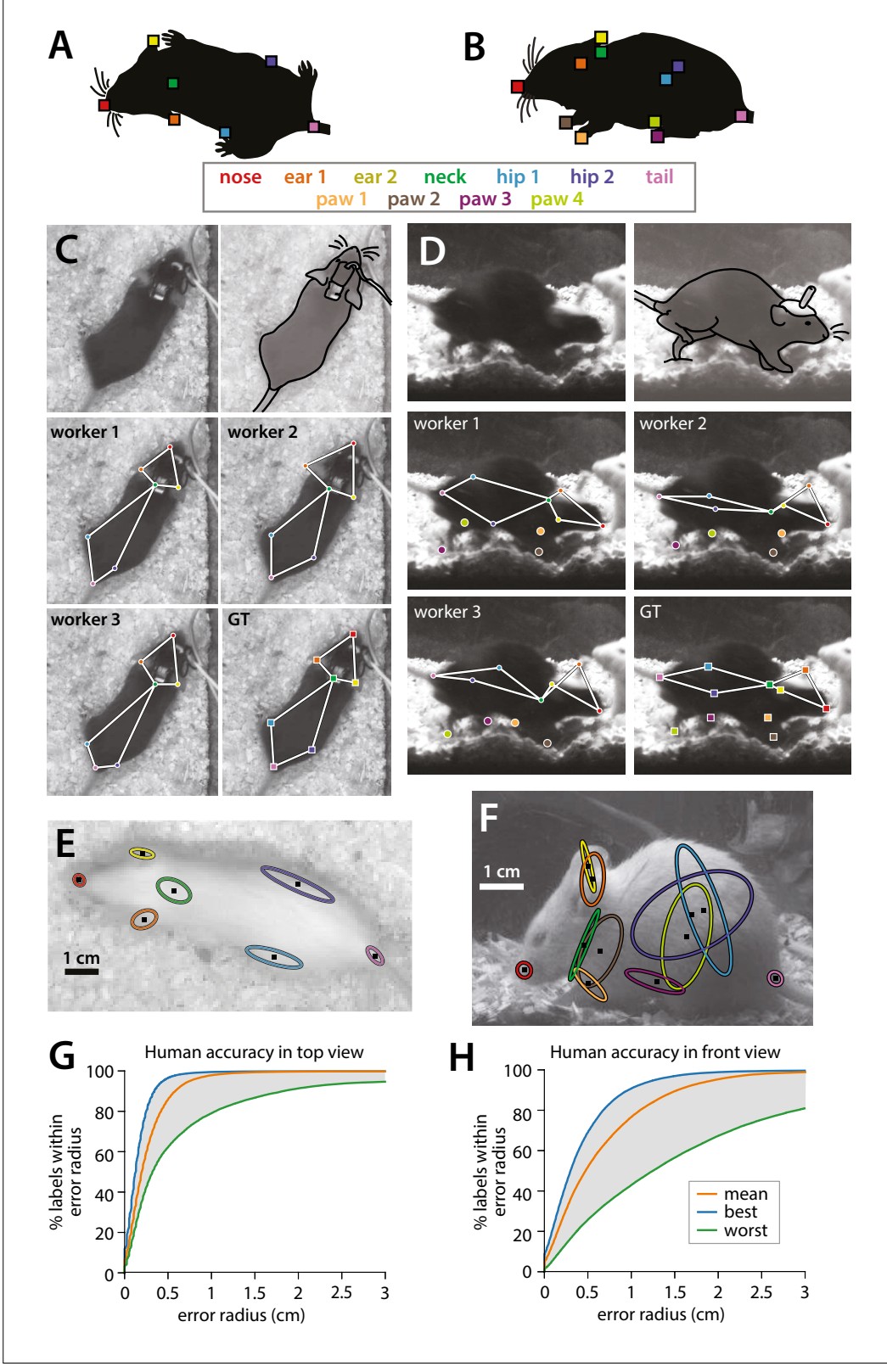

**Figure 2.** Quantifying human annotation variability in top- and front-view pose estimates. (**A, B**) Anatomical keypoints labeled by human annotators in (**A**) top-view and (**B**) front-view movie frames. (**C, D**) Comparison of annotator labels in (**C**) top-view and (**D**) front-view frames. Top row: left, crop of original image shown to annotators (annotators were always provided with the full video frame), right, approximate figure of the mouse (traced for

*Figure 2 continued on next page*

*Figure 2 continued*

clarity). Middle-bottom rows: keypoint locations provided by three example annotators, and the extracted 'ground truth' from the median of all annotations. (**E, F**) Ellipses showing variability of human annotations of each keypoint in one example frame from (**E**) top view and (**F**) front view (N = 5 annotators, 1 standard deviation ellipse radius). (**G, H**) Variability in human annotations of mouse pose for the top-view video, plotted as the percentage of human annotations falling within radius X of ground truth for (**G**) top-view and (**H**) front-view frames.

out in two stages: MARS first detects the body of each mouse (*Figure 3*), then crops the video frame to the detected bounding box and estimates the animal's pose within the cropped image (*Figure 4*).

MARS's detector performs MSC-MultiBox detection (*Szegedy et al., 2014*) using the Inception ResNet v2 (*Szegedy et al., 2017*) network architecture (see Materials and methods for details; *Figure 3A*). We evaluated detection performance using the intersection over union (IoU) metric (*Lin et al., 2014*) for both top- and front-view datasets (*Figure 3B and D*). Plotting precision-recall (PR) curves for various IoU cutoffs revealed a single optimal performance point for both the black and white mice, in both the top and front views (seen as an 'elbow' in plotted PR curves, *Figure 3C and E*).

Following detection, MARS estimates the pose of each mouse using a stacked hourglass network architecture with eight hourglass subunits (*Newell et al., 2016*; *Figure 4A*). Stacked hourglass networks achieve high performance in human pose estimation, and similar two-hourglass architectures produce accurate pose estimates in single-animal settings (*Pereira et al., 2019*; *Graving et al.,*

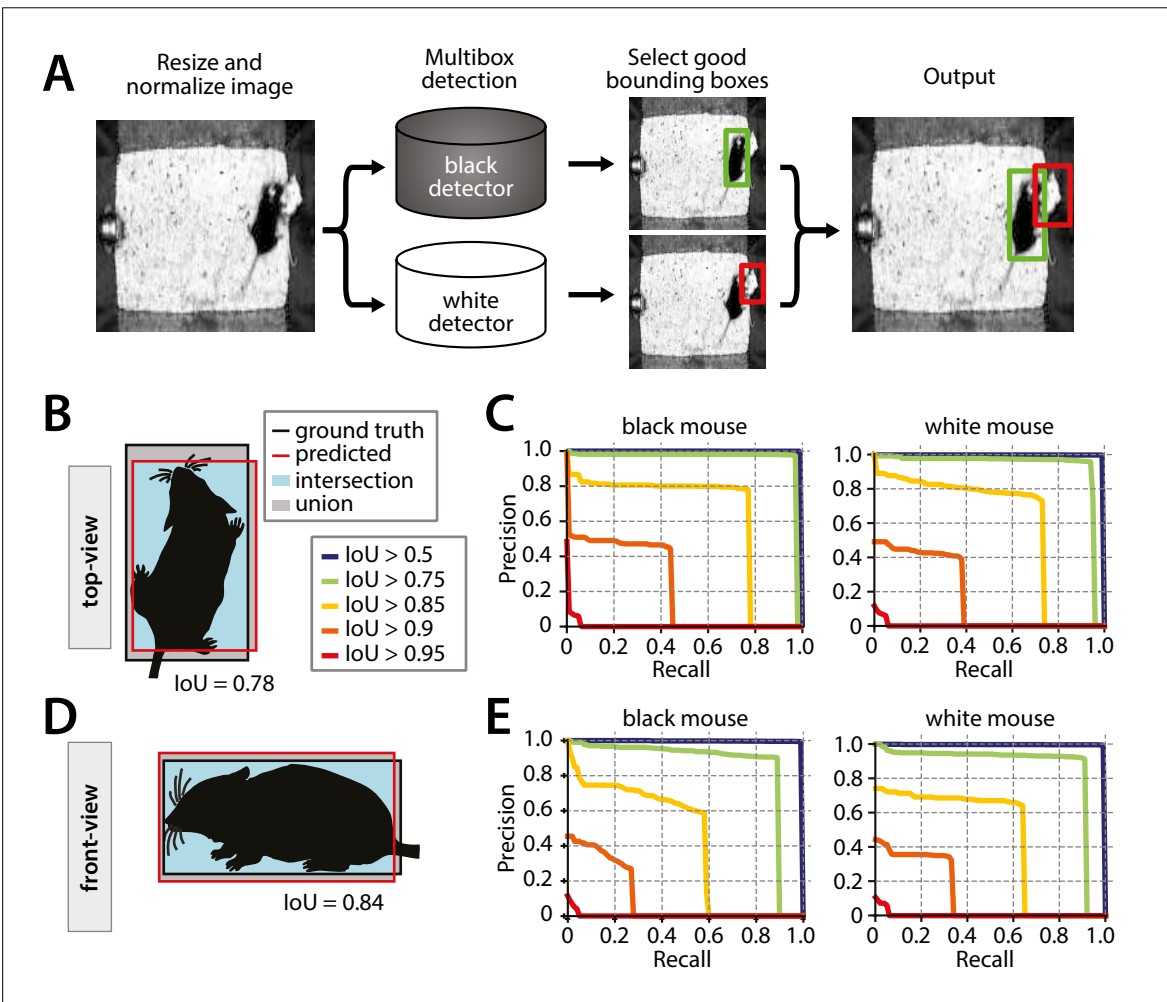

**Figure 3.** Performance of the mouse detection network. (**A**) Processing stages of mouse detection pipeline. (**B**) Illustration of intersection over union (IoU) metric for the top-view video. (**C**) Precision-recall (PR) curves for multiple IoU thresholds for detection of the two mice in the top-view video. (**D**) Illustration of IoU for the front-view video. (**E**) PR curves for multiple IoU thresholds in the front-view video.

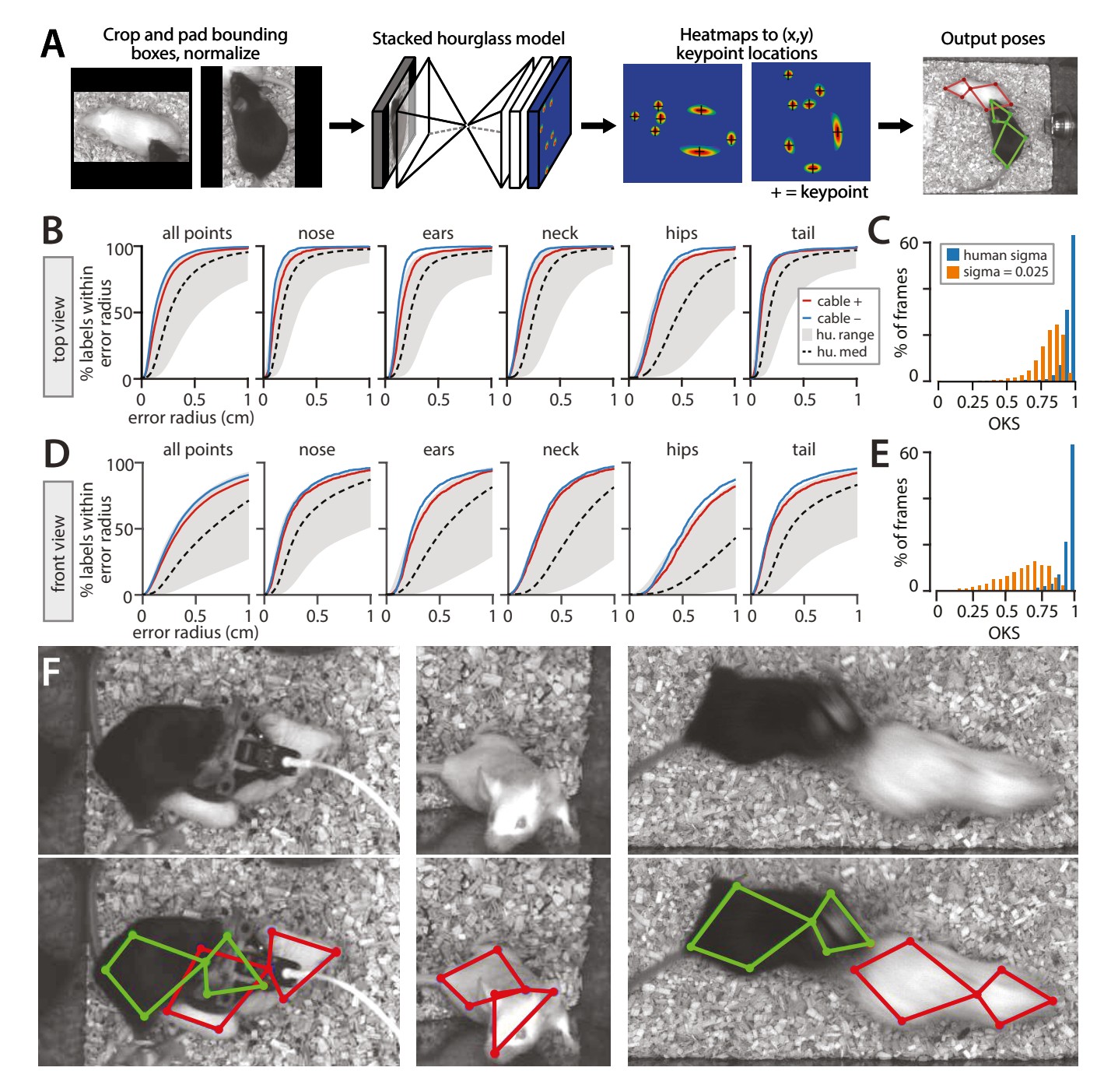

**Figure 4.** Performance of the stacked hourglass network for pose estimation. (**A**) Processing stages of pose estimation pipeline. (**B**) Mouse Action Recognition System (MARS) accuracy for individual body parts, showing performance for videos with vs. without a head-mounted microendoscope or fiber photometry cable on the black mouse. Gray envelop shows the accuracy of the best vs. worst human annotations; dashed black line is median human accuracy. (**C**) Histogram of object keypoint similarity (OKS) scores across frames in the test set. Blue bars: normalized by human annotation variability; orange bars, normalized using a fixed variability of 0.025 (see Materials and methods). (**D**) MARS accuracy for individual body parts in front-view videos with vs. without microendoscope or fiber photometry cables. (**E**) Histogram of OKS scores for the front-view camera. (**F**) Sample video frames (above) and MARS pose estimates (below) in cases of occlusion and motion blur.

The online version of this article includes the following figure supplement(s) for figure 4:

**Figure supplement 1.** Breakdown of Mouse Action Recognition System (MARS) keypoint errors for top- and front-view pose models.

**Table 1.** Performance of MARS top-view pose estimation model.
'Sigma from data' column normalizes pose model performance by observed inter-human variability of each keypoint estimate.

| | Sigma from data | Sigma = 0.025 |
|---|---|---|
| mAP | 0.902 | 0.628 |
| AP@IoU = 50 | 0.99 | 0.967 |
| AP@IoU = 75 | 0.957 | 0.732 |
| mAR | 0.924 | 0.681 |
| AR@IoU = 50 | 0.991 | 0.97 |
| AR@IoU = 75 | 0.97 | 0.79 |

mAP: mean average precision; AP: average precision; mAR: mean average recall; IoU: intersection over union; MARS: Mouse Action Recognition System.

*2019*). The stacked hourglass network architecture pools information across multiple spatial scales of the image to infer the location of keypoints, producing high-quality pose estimates even in cases of occlusion and motion blur (*Figure 4D*).

To contrast MARS performance to human annotator variability, we first evaluated MARS in terms of the PCK metric introduced in *Figure 2*. MARS pose estimates reach the upper limit of human accuracy for both top- and front-view frames, suggesting that the quality of human pose annotation is a limiting factor of the model's performance (*Figure 4B and C*). In the top view, 92% of estimated keypoints fell within 5 mm of ground truth, while in the front view 67% of estimates fell within a 5 mm radius of ground truth (scale bar in *Figure 2E*). Because of the poor performance of front-view pose estimation, we opted to use only the top-view video and pose in our supervised behavior classifiers.

To summarize the performance of MARS's keypoint estimation model, we also used the OKS metric (*Ruggero Ronchi and Perona, 2017*) as this is widely used in the human pose estimation literature (*Lin et al., 2014*; *Xiao et al., 2018*) and has been adopted by other multi-animal pose estimation tools (*Pereira et al., 2019*). For each body part, OKS computes the distance between ground-truth and predicted keypoints, normalized by the variance *sigma* of human annotators labeling that part. We computed the human variance term sigma from our 15,000-frame manual annotation dataset and observed values ranging from 0.039 to 0.084 in top-view images and 0.087–0.125 in front-view images (see Materials and methods). We also computed OKS using a fixed sigma of 0.025 for all body parts for direct comparison with OKS scores reported by SLEAP (*Pereira et al., 2020b*). Following the approach established by COCO (*Lin et al., 2014*), we report OKS in terms of the mean average precision (mAP) and mean average recall (mAR), as well as by the average precision and average recall at two specific IoU cutoffs (see Materials and methods for details). Results are shown in *Table 1*. Finally, we use the approach of Ronchi and Perona to break down keypoint location errors by class (*Figure 4—figure supplement 1*); we find that keypoint error is best accounted for by noise in point placement and by left/right inversion in the front-view pose estimates.

On a desktop computer with 8-core Intel Xeon CPU, 24 Gb RAM, and a 12 GB Titan XP GPU, MARS performs two-animal detection and pose estimation (a total of four operations) at approximately 11 Hz.

## Quantifying inter-annotator variability in the scoring of social behaviors

As in pose estimation, different human annotators can show substantial variability in their annotation of animals' social behaviors, even when those individuals are trained in the same lab. To better understand the variability of human behavioral annotations, we collected annotation data from eight experienced annotators on a common set of 10 behavior videos. Human annotators included three senior laboratory technicians, two postdocs with experience studying mouse social behavior, two graduate students with experience studying mouse social behavior, and one graduate student with previous experience studying fly social behavior. All annotators were instructed to score the same three social behaviors: close investigation, mounting, and attack, and given written descriptions of each behavior (see Materials and methods). Two of the eight annotators showed a markedly different

annotation 'style' with far longer bouts of some behaviors and were omitted from further analysis (see *Figure 5—figure supplement 1*).

We noted several forms of inter-annotator disagreement, consisting of (1) the precise timing of initiation of behavior (*Figure 5—figure supplement 2*), (2) at what point investigation behavior transitioned to attack, and (3) the extent to which annotators merged together multiple consecutive bouts of the same behavior (*Figure 5A*). Other inter-annotator differences that we could not characterize could be ascribed to random variation. Inter-annotator differences in behavior quantification were more pronounced when behavior was reported in terms of total bouts rather than cumulative behavior time, particularly for the two omitted annotators (*Figure 5B and C*, *Figure 5—figure supplement 1*).

Importantly, the precision and recall of annotators was highly dependent on the dataset used for evaluation; we found that the mean annotator F1 score was well predicted by the mean bout duration of annotations in a video, with shorter bout durations leading to lower annotator F1 scores (*Figure 5—figure supplement 3*). This suggests that annotator disagreement over the start and stop times of behavior bouts may be a primary form of inter-annotator variability. Furthermore, this finding shows the importance of standardized datasets for evaluating the performance of different automated annotation approaches.

Finally, to evaluate the stability of human annotations, all eight annotators re-scored 2 of the 10 behavior videos a minimum of 10 months later. We then computed with- vs. between-annotator agreement in terms of F1 score of annotations on these two videos. For both attack and close investigations, within-annotator F1 score was significantly higher than between-annotator F1 score (*Figure 5—figure supplement 4*, full stats in *Table 2*). We hypothesize that this effect was not observed for mounting due to the higher within-annotator agreement for that behavior. These findings support our conclusion that inter-annotator variability reflects a genuine difference in annotation style between individuals, rather than inter-annotator variability being due to noise alone.

## MARS achieves high accuracy in the automated classification of three social behaviors

To create a training set for automatic detection of social behaviors in the resident-intruder assay, we collected manually annotated videos of social interactions between a male resident (black mouse) and a male or female intruder (white mouse). We found that classifiers trained with multiple annotators' labels of the same actions were less accurate than classifiers trained on a smaller pool of annotations from a single individual. Therefore, we trained classifiers on 6.95 hr of video annotated by a single individual (human #1 in *Figure 5*) for attack, mounting, and close investigation behavior. To avoid overfitting, we implemented early stopping of training based on performance on a separate validation set of videos, 3.85 hr in duration. Distributions of annotated behaviors in the training, evaluation, and test sets are reported in *Figure 1—figure supplement 2*.

As input to behavior classifiers, we designed a set of 270 custom spatiotemporal features from the tracked poses of the two mice in the top-view video (full list of features in *Table 3*). For each feature, we then computed the feature mean, standard deviation, minimum, and maximum over windows of 0.1, 0.37, and 0.7 s to capture how features evolved in time. We trained a set of binary supervised classifiers to detect each behavior of interest using the XGBoost algorithm (*Chen and Guestrin, 2016*), then smoothed classifier output and enforced one-hot labeling (i.e., one behavior/frame only) of behaviors with a Hidden Markov Model (HMM) (*Figure 6A*).

When tested on the 10 videos previously scored by multiple human annotators ('test set 1,' 1.7 hr of video, behavior breakdown in *Figure 1—figure supplement 2*), precision and recall of MARS classifiers was comparable to that of human annotators for both attack and close investigation, and slightly below human performance for mounting (*Figure 6B and C*, humans and MARS both evaluated with respect to human #1; *Video 1*). Varying the threshold of a given binary classifier in MARS produces a precision-recall curve (PR curve) showing the tradeoff between the classifier's true-positive rate and its false-positive rate (*Figure 6B–D*, black lines). Interestingly, the precision and recall scores of different human annotators often fell quite close to this PR curve.

False positive/negatives in MARS output could be due to mistakes by MARS; however, they may also reflect noise or errors in the human annotations serving as our 'ground truth.' We therefore also computed the precision and recall of MARS output relative to the pooled (median) labels of all six annotators. To pool annotators, we scored a given frame as positive for a behavior if at least three

**Table 2.** Statistical significance testing.

All t-tests are two-sided unless otherwise stated. All tests from distinct samples unless otherwise stated. Effect size for two-sample t-test is Cohen's d. Effect size for rank sum test is $U/(n_1 * n_2)$, where $n_1$ and $n_2$ are sample sizes of the two categories.

| Figure | Panel | Identifier | Sample size | Statistical test | Test stat. | CI | Effect size | DF | p-Value |
|---|---|---|---|---|---|---|---|---|---|
| | | Chd8 GH mutant vs. GH control | 8 het 8 wt | Two-sample t-test | 2.31 | 0.216–5.85 | 1.155 | 14 | 0.0367 |
| | b | Nlgn3 GH mutant vs. SH mutant | 10 GH 8 SH | Two-sample t-test | 4.40 | 2.79–7.99 | 1.958 | 16 | 0.000449 |
| | c | BTBR SH mutant vs. SH control | 10 het 10 wt | Two-sample t-test | 2.59 | 0.923–8.91 | 1.157 | 18 | 0.0186 |
| | | Close investigate | 10 het 10 wt | Two-sample t-test | 4.58 | 0.276–0.743 | 2.05 | 18 | 0.000230 |
| | | Face-directed | 10 het 10 wt | Two-sample t-test | 3.84 | 0.171–0.582 | 1.72 | 18 | 0.00120 |
| 8 | e | Genital-directed | 10 het 10 wt | Two-sample t-test | 5.01 | 0.233–0.568 | 2.24 | 18 | 0.0000903 |
| | | Attack | 6 vs. self 15 vs. other | Wilcoxon rank sum | U = 79 | × | 0.878 | × | 0.00623 |
| | b | Close investigation | 6 vs. self 15 vs. other | Wilcoxon rank sum | U = 73 | × | 0.811 | × | 0.0292 |
| | | Attack | 8 vs. self 28 vs. other | Wilcoxon rank sum | U = 204 | × | 0.911 | × | 0.000498 |
| ED 8 | d | Close investigation | 8 vs. self 28 vs. other | Wilcoxon rank sum | U = 193 | × | 0.862 | × | 0.00219 |

GH: group-housed; SH: singly housed.

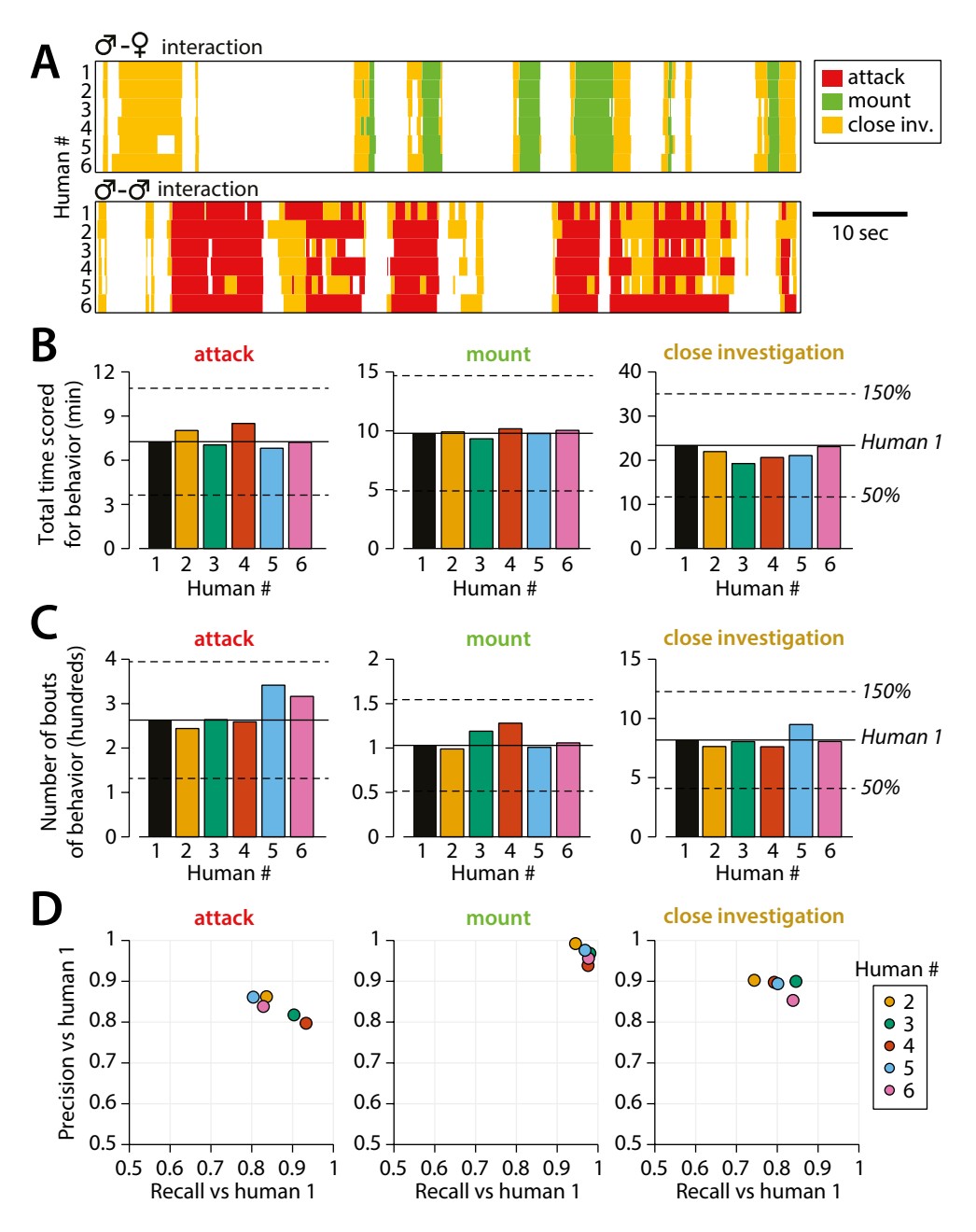

**Figure 5.** Quantifying inter-annotator variability in behavior annotations. (**A**) Example annotation for attack, mounting, and close investigation behaviors by six trained annotators on segments of male-female (top) and male-male (bottom) interactions. (**B**) Inter-annotator variability in the total reported time mice spent engaging in each behavior. (**C**) Inter-annotator variability in the number of reported bouts (contiguous sequences of frames) scored for each behavior. (**D**) Precision and recall of annotators (humans) 2–6 with respect to annotations by human 1.

The online version of this article includes the following figure supplement(s) for figure 5:

**Figure supplement 1.** Expanded set of human annotations.

**Figure supplement 2.** Within-annotator bias and variance in annotation of attack start time.

**Figure supplement 3.** Inter-annotator accuracy on individual videos.

**Figure supplement 4.** Inter- and intra-annotator variability.

out of six annotators labeled it as such. Precision and recall of MARS relative to this 'denoised ground truth' was further improved, particularly for the attack classifier (*Figure 6D and E*).

**Table 3.** MARS feature definitions.

| Name | Units | Definition | Res. | Intr. |
|------|-------|------------|------|-------|
| **Position Features** | | | | |
| (p)_x, (p)_y | cm | x,y coordinates of each body part, for p in (nose, left ear, right ear, neck, left hip, right hip, tail) | x | x |
| centroid_x, centroid_y | cm | x,y coordinates of the centroid of an ellipse fit to the seven keypoints representing the mouse's pose. | x | x |
| centroid_head_x, centroid_head_y | cm | x,y coordinates of the centroid of an ellipse fit to the nose, left and right ear, and neck keypoints. | x | x |
| centroid_hips_x, centroid_hips_y | cm | x,y coordinates of the centroid of an ellipse fit to the left and right hip and tail base keypoints. | x | x |
| centroid_body_x, centroid_body_y | cm | x,y coordinates of the centroid of an ellipse fit to the neck, left and right hip, and tail base keypoints. | x | x |
| dist_edge_x, dist_edge_y | cm | distance from the centroid of the mouse to the closest vertical (dist_edge_x) or horizontal (dist_edge_y) wall of the home cage. | x | x |
| dist_edge | cm | distance from the centroid of the mouse to the closest of the four walls of the home cage. | x | x |
| **Appearance Features** | | | | |
| phi | radians | absolute orientation of the mouse, measured by the orientation of a vector from the centroid of the head to the centroid of the hips. | x | x |
| ori_head | radians | absolute orientation of a vector from the neck to the tip of the nose. | x | x |
| ori_body | radians | absolute orientation of a vector from the tail to the tip of the neck. | x | x |
| angle_head_body_l, angle_head_body_r | radians | angle formed by the left(right) ear, neck and left(right) hip keypoints. | x | x |
| major_axis_len, minor_axis_len | cm | major and minor axis of an ellipse fit to the seven keypoints representing the mouse's pose. | x | x |
| axis_ratio | none | major_axis_len/minor_axis_len (as defined above). | x | x |
| area_ellipse | cm^2 | area of the ellipse fit to the mouse's pose. | x | x |
| dist_(p1)(p2) | cm | distance between all pairs of keypoints (p1, p2) of the mouse's pose. | x | x |
| **Locomotion Features** | | | | |
| speed | cm/s | mean change in position of centroids of the head and hips (see Position Features), computed across two consecutive frames. | x | x |
| speed_centroid | cm/s | change in position of the mouse's centroid (see Position Features), computed across two consecutive frames. | x | x |
| acceleration | cm/s^2 | mean change in speed of centroids of the head and hips, computed across two consecutive frames. | x | x |
| acceleration_centroid | cm/s^2 | change in speed of the mouse's centroid, computed across two consecutive frames. | x | x |
| speed_fwd | cm/s | speed of the mouse in the direction of ori_body (see Appearance Features). | x | x |
| radial_vel | cm/s | component of the mouse's centroid velocity along the vector between the centroids of the two mice, computed across two consecutive frames. | x | x |
| tangential_vel | cm/s | component of the mouse's centroid velocity tangential to the vector between the centroids of the two mice, computed across two consecutive frames. | x | x |

*Table 3 continued on next page*

*Table 3 continued*

| Name | Units | Definition | Res. | Intr. |
|---|---|---|---|---|
| speed_centroid_w(s) | cm/s | speed of the mouse's centroid, computed as the change in position between timepoints (s) frames apart (at 30 Hz). | x | x |
| speed_(p)_w(s) | cm/s | speed of each keypoint (p) of the mouse's pose, computed as the change in position between timepoints (s) frames apart (at 30 Hz). | x | x |
| Image-based features | | | | |
| pixel_change | none | mean squared value of (pixel intensity on current frame minus mean pixel intensity on previous frame) over all pixels, divided by mean pixel intensity on current frame (as defined in [Hong et al].) | x | |
| pixel_change_ubbox_ mice | none | pixel change (as above) computed only on pixels within the union of the bounding boxes of the detected mice (when bounding box overlap is greater than 0; 0 otherwise). | x | |
| (p)_pc | none | pixel change (as above) within a 20 pixel-diameter square around the keypoint for each body part (p). | x | x |
| Social features | | | | |
| resh_twd_itrhb (resident head toward intruder head/body) | none | binary variable that is one if the centroid of the other mouse is within a –45° to 45° cone in front of the animal. | x | x |
| rel_angle_social | radians | relative angle between the body of the mouse (ori_body) and the line connecting the centroids of both mice. | x | x |
| rel_dist_centroid | cm | distance between the centroids of the two mice. | x | |
| rel_dist_centroid_change | cm | change in distance between the centroids of the two mice, computed across two consecutive frames. | x | |
| rel_dist_gap | cm | distance between ellipses fit two the two mice along the vector between the two ellipse centers, equivalent to Feature 13 of [Hong et al]. | x | |
| rel_dist_scaled | cm | distance between the two animals along the line connecting the two centroids, divided by length of the major axis of one mouse, equivalent to Feature 14 of [Hong et al]. | x | x |
| rel_dist_head | cm | distance between centroids of ellipses fit to the heads of the two mice. | x | |
| rel_dist_body | cm | distance between centroids of ellipses fit to the bodies of the two mice. | x | |
| rel_dist_head_body | cm | distance from the centroid of an ellipse fit to the head of mouse A to the centroid of an ellipse fit to the body of mouse B. | x | x |
| overlap_bboxes | none | intersection over union of the bounding boxes of the two mice. | x | |
| area_ellipse_ratio | none | ratio of the areas of ellipses fit to the poses of the two mice. | x | x |
| angle_between | radians | angle between mice defined as the angle between the projection of the centroids. | x | |
| facing_angle | radians | angle between head orientation of one mouse and the line connecting the centroids of both animals. | x | x |
| dist_m1(p1)_m2(p2) | cm | distance between keypoints of one mouse w.r.t to the other, for all pairs of keypoints (p1, p2). | x | |

MARS: Mouse Action Recognition System.

The precision and recall of MARS on individual videos was highly correlated with the average precision and recall of individual annotators with respect to the annotator median (*Figure 6—figure supplement 1*). Hence, as for human annotators, precision and recall of MARS are correlated with the

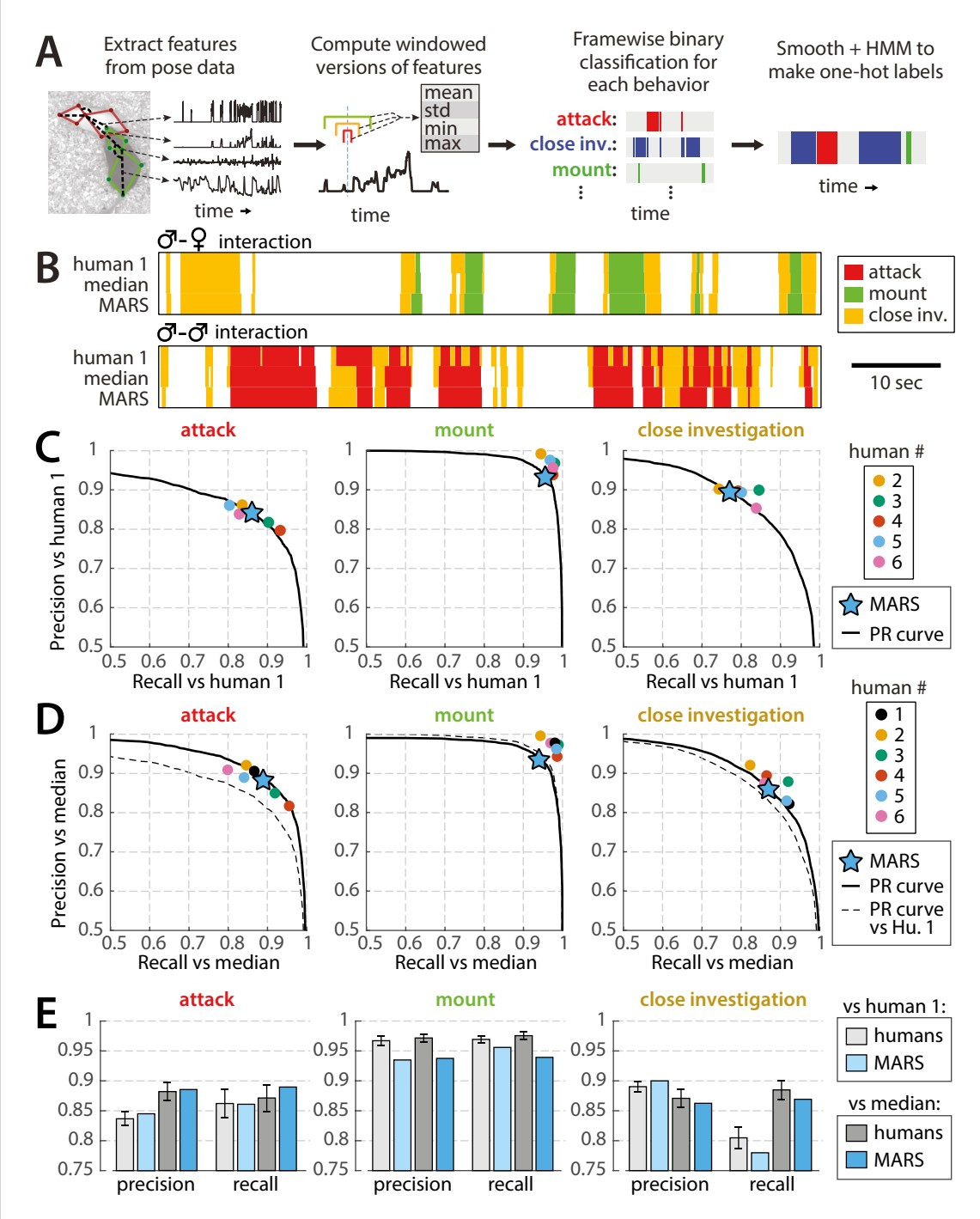

**Figure 6.** Performance of behavior classifiers. (**A**) Processing stages of estimating behavior from pose of both mice. (**B**) Example output of the Mouse Action Recognition System (MARS) behavior classifiers on segments of male-female and male-male interactions compared to annotations by human 1 (source of classifier training data) and to the median of the six human annotators analyzed in *Figure 5*. (**C**) Precision, recall, and precision-recall (PR) curves of MARS with respect to human 1 for each of the three behaviors. (**D**) Precision, recall, and PR curves of MARS with respect to the median of the six human annotators (precision/recall for each human annotator was computed with respect to the median of the other five). (**E**) Mean precision and recall of human annotators vs. MARS, relative to human 1 and relative to the group median (mean ± SEM).

The online version of this article includes the following figure supplement(s) for figure 6:

**Figure supplement 1.** Mouse Action Recognition System (MARS) precision and recall is closely correlated with that of annotators on individual videos.

**Figure supplement 2.** Evaluation of Mouse Action Recognition System (MARS) on a larger test set.

**Figure supplement 3.** Training Mouse Action Recognition System (MARS) on new datasets.

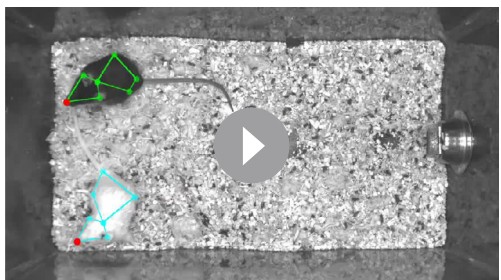

**Video 1.** Sample MARS output.
https://elifesciences.org/articles/63720/figures#video1

average duration of behavior bouts in a video (see *Figure 6—figure supplement 1B*), with longer behavior bouts leading to higher precision and recall values.

We next tested MARS classifiers on a second set of videos of mice with head-mounted micro-endoscopes or other cable-attached devices ('test set 2,' 1.66 hr of video, behavior breakdown in *Figure 1—figure supplement 2*). While precision and recall curves differ on this test set (*Figure 6—figure supplement 2A*), we do not observe a difference on individual videos with vs. without cable when controlling for mean bout length in the video (*Figure 6—figure supplement 2B*). We therefore conclude that MARS classifier performance is robust to occlusions and motion artifacts produced by head-mounted recording devices and cables.

## Training MARS on new data

While MARS can be used out of the box with no training data, it is often useful to study additional social behaviors or track animals in different environments. The MARS_Developer library allows users to re-train MARS detection and pose estimation models in new settings, and to train their own behavior classifiers from manually annotated videos. The training code in this library is the same code that was used to produce the end-user version of MARS presented in this paper.

To demonstrate the functionality of MARS_Developer, we trained detection and pose estimation networks on the CRIM13 dataset (*Burgos-Artizzu et al., 2012*). We used the pose_annotation_tools library to crowdsource manual annotation of animal pose on 5,577 video frames, with five workers per frame (cost: \$0.38/image). We then trained detection and pose models using the existing MARS detection and pose models as starting points for training. We found that the performance of trained models improved as a function of training set size, plateauing at around 1500 frames (*Figure 6—figure supplement 3A–C*). We also trained behavior classifiers for three additional social behaviors of interest: face-directed sniffing, anogenital-directed sniffing, and intromission, using a subset of the MARS training set annotated for these behaviors. Trained classifiers achieved F1 scores of at least 0.7 for all three behaviors; by training on subsets of the full MARS dataset, we found that classifier performance improves logarithmically with training set size (*Figure 6—figure supplement 3D and E*). Importantly, the number of annotated bouts of a behavior is a better predictor of classifier performance than the number of annotated frames.

## Integration of video, annotation, and neural recording data in a user interface

Because one objective of MARS is to accelerate the analysis of behavioral and neural recording data, we developed an open-source interface to allow users to more easily navigate neural recording, behavior video, and tracking data (*Figure 7A*). This tool, called the Behavior Ensemble and Neural Trajectory Observatory (BENTO), allows users to synchronously display, navigate, analyze, and save movies from multiple behavior videos, behavior annotations, MARS pose estimates and features, audio recordings, and recorded neural activity (*Video 2*). BENTO is currently MATLAB-based, although a Python version is in development.

BENTO includes an interface for manual annotation of behavior, which can be combined with MARS to train, test, and apply classifiers for novel behaviors. A button in the BENTO interface allows users to launch training of new MARS behavior classifiers directly from their annotations. BENTO also allows users to access MARS pose features directly to create handcrafted filters on behavioral data (*Figure 7B*). For example, users may create and apply a filter on inter-animal distance or resident velocity to automatically identify all frames in which feature values fall within a specified range.

BENTO also provides interfaces for several common analyses of neural activity, including event-triggered averaging, 2D linear projection of neural activity, and clustering of cells by their activity.

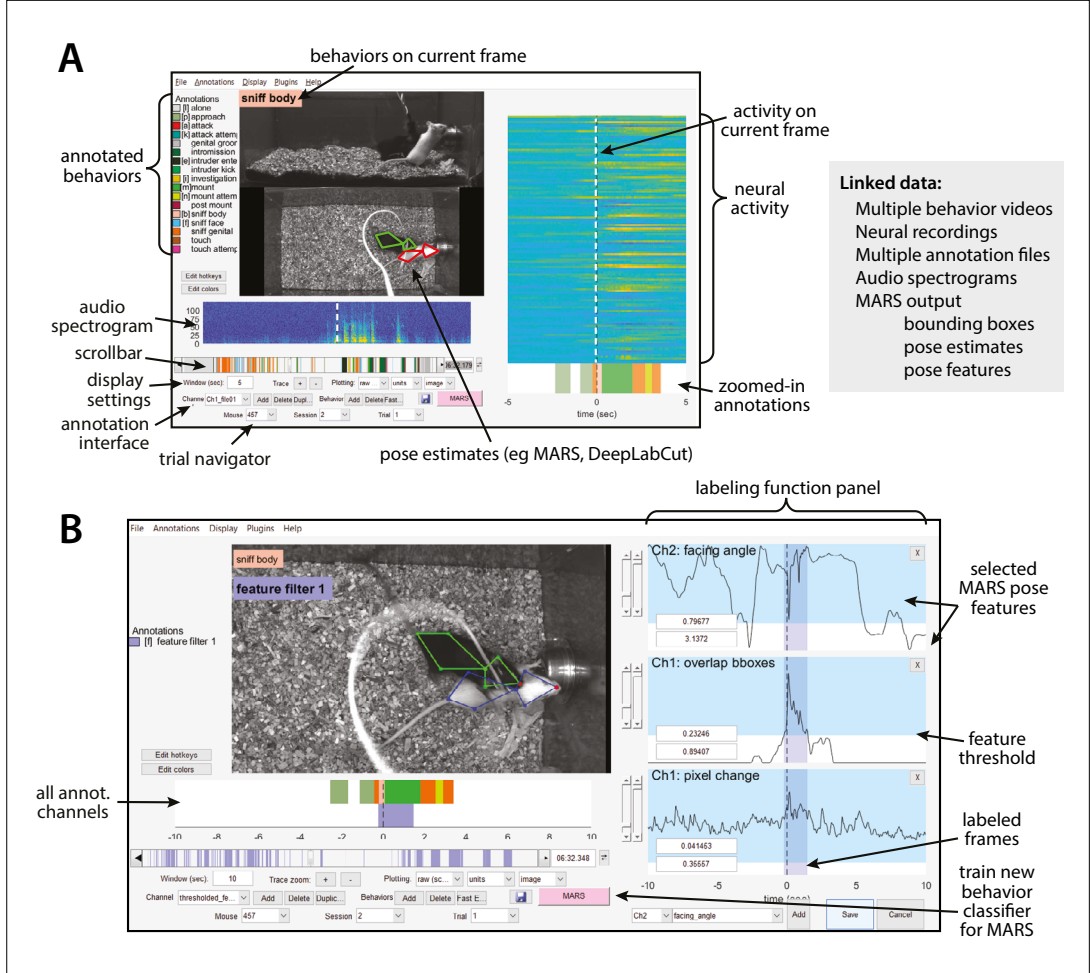

**Figure 7.** Screenshot of the Behavior Ensemble and Neural Trajectory Observatory (BENTO) user interface. (**A**, left) The main user interface showing synchronous display of video, pose estimation, neural activity, and pose feature data. (Right) List of data types that can be loaded and synchronously displayed within BENTO. (**B**) BENTO interface for creating annotations based on thresholded combinations of Mouse Action Recognition System (MARS) pose features.

Advanced users have the option to create additional custom analyses as plug-ins in the interface. BENTO is freely available on GitHub and is supported by documentation and a user wiki.

## Use case 1: high-throughput social behavioral profiling of multiple genetic mouse model lines

Advances in human genetics, such as genome-wide association studies (GWAS), have led to the identification of multiple gene loci that may increase susceptibility to autism (*Grove et al., 2019*; *O'Roak et al., 2012*). The laboratory mouse has been used as a system for studying 'genocopies' of allelic variants found in humans, and dozens of mouse lines containing engineered autism-associated mutations have been produced in an effort to understand the effect of these mutations on neural circuit development and function (*Silverman et al., 2010*, *Moy and Nadler, 2008*). While several lines show atypical social behaviors, it is unclear whether all lines share a similar

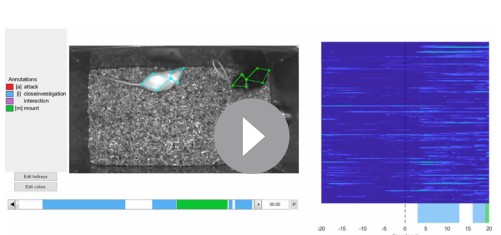

**Video 2.** Joint display of video, pose estimates, neural activity, and behavior within BENTO.
https://elifesciences.org/articles/63720/figures#video2

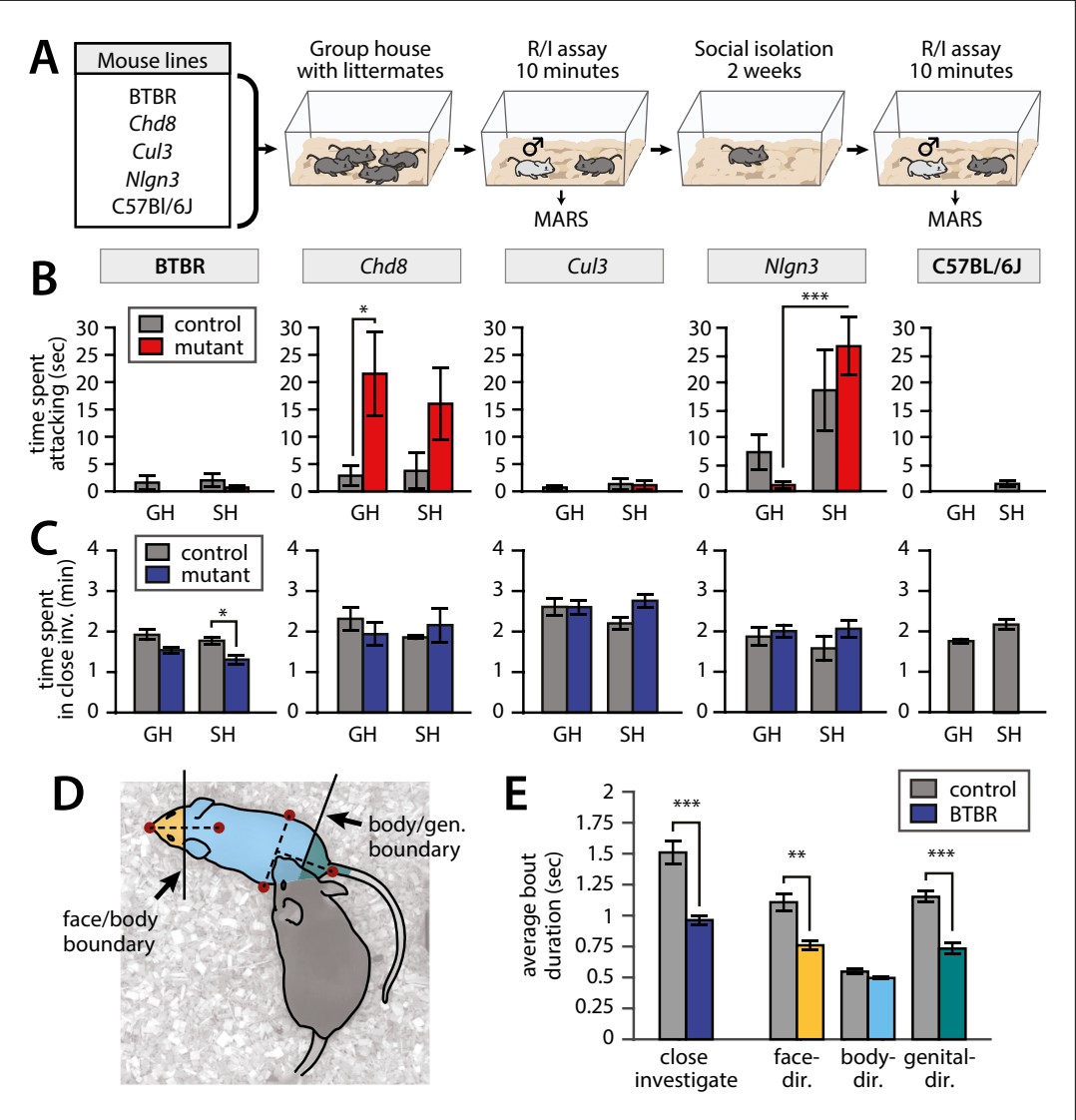

**Figure 8.** Application of Mouse Action Recognition System (MARS) in a large-scale behavioral assay. All plots: mean ± SEM, N = 8–10 mice per genotype per line (83 mice total); *p<0.05, **p<0.01, ***p<0.001. (**A**) Assay design. (**B**) Time spent attacking by group-housed (GH) and single-housed (SH) mice from each line compared to controls (*Chd8* GH het vs. ctrl: p=0.0367, Cohen's d = 1.155, two-sample t-test, N = 8 het vs. 8 ctrl; *Nlgn3* het GH vs. SH: p=0.000449, Cohen's d = 1.958, two-sample t-test, N = 10 GH vs. 8 SH). (**C**) Time spent engaged in close investigation by each condition/line (BTBR SH BTBR vs. ctrl: p=0.0186, Cohen's d = 1.157, two-sample t-test, N = 10 BTBR vs. 10 ctrl). (**D**) Cartoon showing segmentation of close investigation bouts into face-, body-, and genital-directed investigation. Frames are classified based on the position of the resident's nose relative to a boundary midway between the intruder mouse's nose and neck, and a boundary midway between the intruder mouse's hips and tail base. (**E**) Average duration of close investigation bouts in BTBR mice for investigation as a whole and broken down by the body part investigated (close investigation, p=0.00023, Cohen's d = 2.05; face-directed p=0.00120, Cohen's d = 1.72; genital-directed p=0.0000903, Cohen's d = 2.24; two-sample t-test, N = 10 het vs. 10 ctrl for all).

behavioral profile or whether different behavioral phenotypes are associated with different genetic mutations.

We collected and analyzed a 45 hr dataset of male-male social interactions using mice from five different lines: three lines that genocopy autism-associated mutations (Chd8 [*Katayama et al., 2016*], Cul3 [*Dong et al., 2020*], and Nlgn3 [*Tabuchi et al., 2007*]), one inbred line that has previously been shown to exhibit atypical social behavior and is used as an autism 'model' (BTBR [*Hong et al., 2015*]),

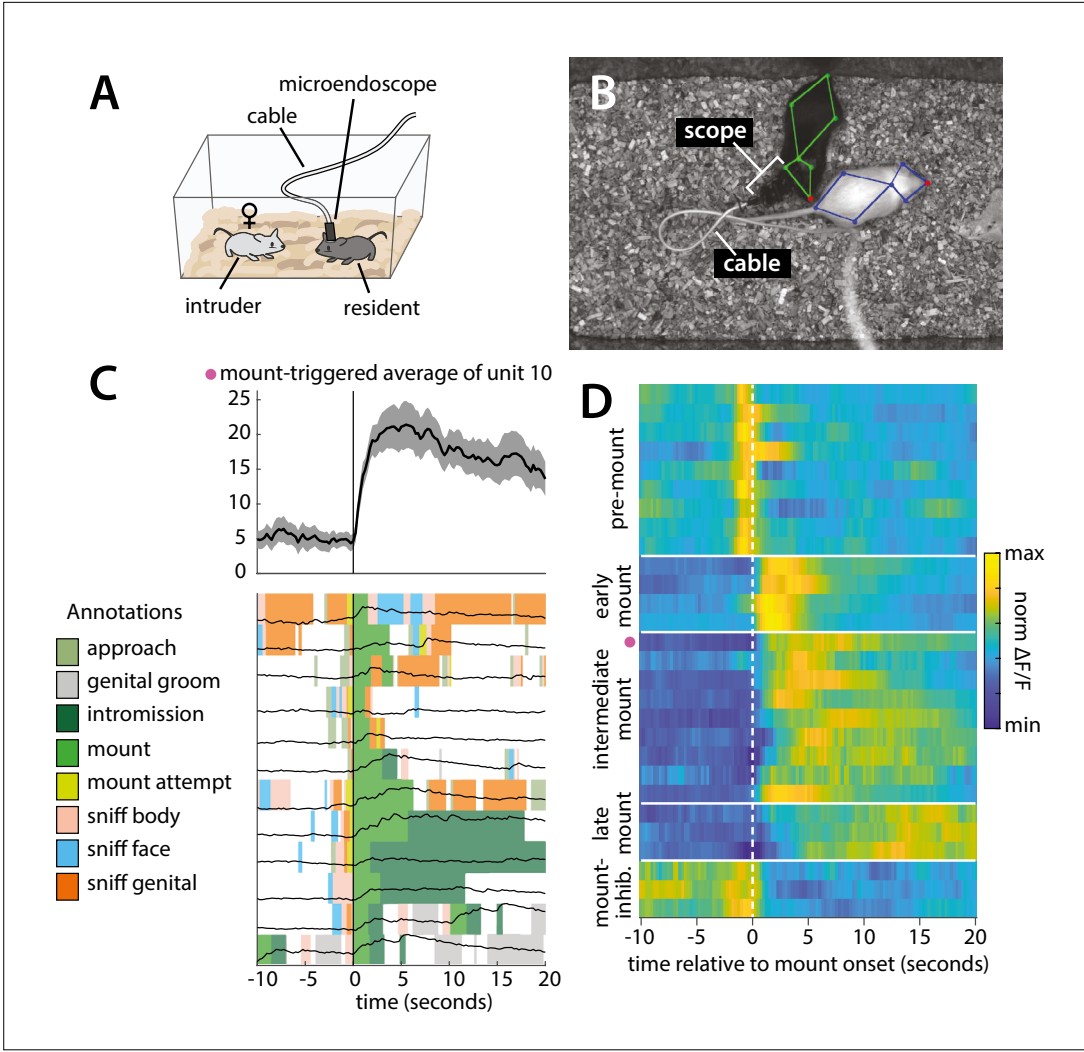

**Figure 9.** Analysis of a microendoscopic imaging dataset using Mouse Action Recognition System (MARS) and Behavior Ensemble and Neural Trajectory Observatory (BENTO). (**A**) Schematic of the imaging setup, showing head-mounted microendoscope. (**B**) Sample video frame with MARS pose estimate, showing appearance of the microendoscope and cable during recording. (**C**) Sample behavior-triggered average figure produced by BENTO. (Top) Mount-triggered average response of one example neuron within a 30 s window (mean ± SEM). (Bottom) Individual trials contributing to mount-triggered average, showing animal behavior (colored patches) and neuron response (black lines) on each trial. The behavior-triggered average interface allows the user to specify the window considered during averaging (here 10 s before to 20 s after mount initiation), whether to merge behavior bouts occurring less than X seconds apart, whether to trigger on behavior start or end, and whether to normalize individual trials before averaging; results can be saved as a pdf or exported to the MATLAB workspace. (**D**) Normalized mount-triggered average responses of 28 example neurons in the medial preoptic area (MPOA), identified using BENTO. Grouping of neurons reveals diverse subpopulations of cells responding at different times relative to the onset of mounting (pink dot = neuron shown in panel **C**).

and a C57Bl/6J control line. For each line, we collected behavior videos during 10 min social interactions with a male intruder and quantified expression of attack and close investigation behaviors using MARS. In autism lines, we tested heterozygotes vs. age-matched wild-type (WT) littermates; BTBR mice were tested alongside age-matched C57Bl/6J mice. Due to the need for contrasting coat colors to distinguish interacting mouse pairs, all mice were tested with BalbC intruder males. Each mouse was tested using a repeated measures design (*Figure 8A*): first in their home cage after group-housing with heterozygous and WT littermates, and again after 2 weeks of single-housing.

Consistent with previous studies, MARS annotations showed increased aggression in group-housed *Chd8*[+/-] relative to WT littermate controls (*Figure 8B*; full statistical reporting in Table S2). *Nlgn3*[+/-] were

more aggressive than C57Bl/6J animals, consistent with previous work (*Burrows et al., 2015*), and showed a significant increase in aggression following single-housing. But, interestingly, there was not a statistically significant difference in aggression between single-housed *Nlgn3*$^{+/-}$ and their WT littermates, which were also aggressive. The increased aggression of WT littermates of *Nlgn3*$^{+/-}$ may be due to their genetic background (C57Bl6-SV129 hybrid rather than pure C57Bl/6) or could arise from the environmental influence of these mice being co-housed with aggressive heterozygote littermates (*Kalbassi et al., 2017*).

We also confirmed previous findings (*Hong et al., 2015*) that BTBR mice spend less time investigating intruder mice than C57Bl/6J control animals (*Figure 8C*), and that the average duration of close investigation bouts was reduced (*Figure 8E*, left). Using MARS's estimate of the intruder mouse's pose, we defined two anatomical boundaries on the intruder mouse's body: a 'face-body boundary' midway between the nose and neck keypoints, and a 'body-genital boundary' midway between the tail and the center of the hips. We used these boundaries to automatically label frames of close investigation as either face-, body-, or genital-directed investigation (*Figure 8D*). This relabeling revealed that in addition to showing shorter investigation bouts in general, the BTBR mice showed shorter bouts of face- and genital-directed investigation compared to C57Bl/6J controls, while the duration of body-directed investigation bouts was not significantly different from controls (*Figure 8E*). This finding may suggest a loss of preference for investigation of, or sensitivity to, socially relevant pheromonal cues in the BTBR inbred line.

Without automation of behavior annotation by MARS, analysis of body part-specific investigation would have required complete manual reannotation of the dataset, a prohibitively slow process. Our findings in BTBR mice therefore demonstrate the power of MARS behavior labels and pose features as a resource for exploratory analysis of behavioral data.

## Use case 2: finding neural correlates of mounting behavior

The sensitivity of electrophysiological and two-photon neural imaging methods to motion artifacts has historically required the recording of neural activity to be performed in animals that have been head-fixed or otherwise restrained. However, head-fixed animals cannot perform many naturalistic behaviors, including social behaviors. The emergence of novel technologies such as microendoscopic imaging and silicon probe recording has enabled the recording of neural activity in freely moving animals *Resendez et al., 2016*; however, these techniques still require animals to be fitted with a head-mounted recording device, typically tethered to an attached cable (*Figure 9A and B*).

To demonstrate the utility of MARS and BENTO for these data, we analyzed data from a recent study in the Anderson lab (*Karigo et al., 2021*), in which a male Esr1+ Cre mouse was implanted with a microendoscopic imaging device targeting the medial preoptic area (MPOA), a hypothalamic nucleus implicated in social and reproductive behaviors (*Karigo et al., 2021*; *Wei et al., 2018*; *Wu et al., 2014*). We first used MARS to automatically detect bouts of close investigation and mounting while this mouse freely interacted with a female conspecific. Next, video, pose, and annotation data were loaded into BENTO, where additional social behaviors of interest were manually annotated. Finally, we re-loaded video, pose, and annotation data in BENTO alongside traces of neural activity extracted from the MPOA imaging data.

Using BENTO's behavior-triggered average plug-in, we visualized the activity of individual MPOA neurons when the animal initiated mounting behavior (*Figure 9C*) and identified a subset of 28 imaged neurons whose activity was modulated by mounting. Finally, using a subset of these identified cells, we exported mount-averaged activity from the behavior-triggered average plug-in and visualized their activity as a heatmap (*Figure 9D*). This analysis allowed us to quickly browse this imaging dataset and determine that multiple subtypes of mount-modulated neurons exist within the imaged MPOA population, with all analysis except for the final plotting in *Figure 9D* performed from within the BENTO user interface.

## Discussion

Automated systems for accurate pose estimation are increasingly available to neuroscientists and have proven to be useful for measuring animal pose and motion in a number of studies (*Datta et al., 2019*; *Pereira et al., 2020a*; *Mathis and Mathis, 2020*; *Dell et al., 2014*). However, pose alone does not

provide insight into an animal's behavior. Together, MARS and BENTO provide an end-to-end tool for automated pose estimation and supervised social behavior classification in the widely used resident-intruder assay and links these analyses with a GUI for the quick exploration and analysis of joint neural and behavioral datasets. MARS allows users to perform high-throughput screening of social behavior expression and is robust to occlusion and motion from head-mounted recording devices and cables. The pretrained version of MARS does not require users to collect and annotate their own keypoint annotations as training data, and runs out-of-the-box on established hardware (*Hong et al., 2015*). For behavioral data collected in other settings, the MARS_Developer repository allows users to fine-tune MARS's pose estimator and behavior classifiers with their own data. MARS_Developer also allows users to train their own new behavior classifiers from annotations created within BENTO. Future versions of MARS_Developer will incorporate our fast-learning semi-supervised behavior analysis tool Trajectory Embedding for Behavior Analysis (TREBA *Sun et al., 2021b*) to train behavior classifiers in arbitrary settings.

There is to date no field-wide consensus definition of attack, mounting, and investigation behaviors: the classifiers distributed with MARS reflect the careful annotations of one individual in the Anderson lab. MARS's support for classifier re-training allows labs to train MARS on their own annotation data to contrast their 'in-house' definitions of social behaviors of interest to those used in the MARS classifiers. Comparison of trained classifiers may help to identify differences in annotation style between individuals to establish a clearer consensus definition of behaviors (*Sun et al., 2021a*).

MARS operates without requiring multiple cameras or specialized equipment such as a depth camera, unlike our previously published system (*Hong et al., 2015*). MARS is also computationally distinct from our previous work: while Hong et al. used a MATLAB implementation of cascaded pose regression (*Dollár et al., 2010*) to fit an ellipse to the body of each mouse (a form of blob-based tracking), MARS is built using the deep learning package TensorFlow and performs true pose estimation, in that it predicts the location of individual anatomical landmarks on the bodies of the tracked mice. In terms of performance, MARS is also much more accurate and invariant to changes in lighting and to the presence of head-mounted cables compared to our earlier effort. Eliminating the IR-based depth sensor simplifies data acquisition and speeds up processing, and also allows MARS to be used without creating IR artifacts during microendoscopic imaging.

Comparing pose and behavior annotations from multiple human annotators, we were able to quantify the degree of inter-human variability in both tasks and found that in both cases MARS performs comparably to the best-performing human. This suggests that improving the quality of training data, for example, by providing better visualizations and clearer instructions to human annotators, could help to further improve the accuracy of pose estimation and behavior classification tools such as MARS. Conversely, the inter-human variability in behavior annotation may reflect the fact that animal behavior is too complex and heterogeneous for behavior labels of 'attack' and 'close investigation' to be applied consistently by multiple annotators. It is unclear whether future behavior classification efforts with more granular action categories could reduce inter-human variability and lead to higher performance by automated classifiers, or whether more granular categories would cause only greater inter-annotator disagreement, while also requiring more annotation time to collect sufficient labeled training examples for each action.

Unsupervised approaches are a promising alternative to behavior quantification and may eventually bypass the need for human input during behavior discovery (*Wiltschko et al., 2015*; *Berman et al., 2014*; *Vogelstein et al., 2014*; *Luxem et al., 2020*; *Hsu and Yttri, 2021*). While current efforts in unsupervised behavior discovery have largely been limited to single animals, the pose estimates and features produced by MARS could potentially prove useful for future investigations that identify behaviors of interest in a user-unbiased manner. Alternatively, unsupervised analysis of MARS pose features may help to reduce redundancy among features, potentially leading to a reduction in the amount of sample data required to train classifiers to detect new behaviors of interest. We have recently developed one self-supervised tool, called Trajectory Embedding for Behavior Analysis (TREBA), that uses learned embeddings of animal movements to learn more effective features for behavior classification (*Sun et al., 2021b*); support for TREBA feature learning will be incorporated into a future release of MARS_Developer.

Neuroscience has seen an explosive proliferation of tools for automated pose estimation and behavior classification, distributed in accessible open-source packages that have fostered widespread

adoption (*Mathis et al., 2018*, *Graving et al., 2019*; *Nilsson, 2020*; *Pereira et al., 2020b*; *Walter and Couzin, 2021*). However, these tools still require users to provide their own labeled training data, and their performance still depends on training set size and quality. And while the annotation process post-training is made faster and more painless, each lab is still left to create its own definition for each behavior of interest, with no clear mechanisms in place in the community to standardize or compare classifiers. To address these issues, MARS provides a pretrained pose estimation and behavior classification system, with publicly available, well-characterized training data. By this approach, MARS is intended to facilitate comparison of behavioral results between labs, fostering large-scale screening of social behaviors within a common analysis platform.

## Materials and methods
### Data collection
#### Animals
*Chd8*[+/-] and *Cul3*[+/-] mice were obtained from Dr. Mark Zylka, BTBR and *Nlgn3*[+/-] mice were obtained from Jackson Labs (BTBR Stock No. 2282, Nlgn3 Stock No. 8475), and wild-type C57Bl/6J and BALB/c mice were obtained from Charles River. All mice were received at 6–10 weeks of age and were maintained in the Caltech animal facility, where they were housed with three same-sex littermates (unless otherwise noted) on a reverse 11 hr dark 13 hr light cycle with food and water ad libitum. Behavior was tested during the dark cycle. All experimental procedures involving the use of live animals or their tissues were performed in accordance with the NIH guidelines and approved by the Institutional Animal Care and Use Committee (IACUC) and the Institutional Biosafety Committee at the California Institute of Technology (Caltech).

#### The resident-intruder assay
Testing for social behaviors using the resident-intruder assay (*Blanchard et al., 2003*) was performed as in *Zelikowsky et al., 2018*; *Lee et al., 2014*; *Hong et al., 2014*; and *Hong et al., 2015*. Experimental mice ('residents') were transported in their homecage (with cagemates removed) to a behavioral testing room and acclimatized for 5–15 min. Homecages were then inserted into a custom-built hardware setup (*Hong et al., 2015*) with infrared video captured at 30 fps from top- and front-view cameras (Point Grey Grasshopper3) recorded at 1024 × 570 (top) and 1280 × 500 (front) pixel resolution using StreamPix video software (NorPix). Following two further minutes of acclimatization, an unfamiliar group-housed male or female BALB/c mouse ('intruder') was introduced to the cage, and animals were allowed to freely interact for a period of approximately 10 min. BALB/c mice are used as intruders for their white coat color (simplifying identity tracking), as well as their relatively submissive behavior, which reduces the likelihood of intruder-initiated aggression. Social behaviors were manually scored on a frame-by-frame basis, as described in the 'Mouse behavior annotation' section below.

Videos for training of MARS detector, pose estimator, and behavior classifier were selected from previously performed experiments in the Anderson lab. In approximately half of the videos in the training data, mice were implanted with a cranial cannula or with a head-mounted miniaturized microscope (nVista, Inscopix) or optical fiber for optogenetics or fiber photometry, attached to a cable of varying color and thickness. Surgical procedures for these implantations can be found in *Karigo et al., 2021* and *Remedios et al., 2017*.

#### Screening of autism-associated mutation lines
Group-housed heterozygous male mice from mouse lines with autism-associated mutations (*Chd8*[+/-], *Cul3*[+/-], BTBR, and *Nlgn3*[+/-], plus C57Bl/6J control mice) were first tested in a standard resident-intruder assay as outlined above. To control for differences in rearing between lines, an equal number of wild-type littermate male mice from each line were tested on the same day; for the inbred BTBR strain, C57Bl/6J mice were used as controls. Between 8 and 10 animals were tested from each condition, alongside an equal number of controls (*Chd8*[+/-]: 10 het + 10 control; *Cul3*[+/-]: 8 het + 7 control; BTBR: 10 mice + 10 C57Bl/6J control; *Nlgn3*[+/-]: 10 het + 9 control; C57Bl/6J 12 mice). This sample size range was chosen to be in line with similar resident-intruder experiments in the Anderson lab.

*Nlgn3*[+/-] (plus wild-type littermate controls), BTBR, and C57Bl/6J were tested at 11–13 weeks of age; *Cul3*[+/-] and *Chd8*[+/-] mice (plus wild-type littermate controls) were tested at 35–50 weeks of age

as previous studies had noted no clear aggression phenotype in younger animals (*Katayama et al., 2016*). Following the resident-intruder assay, mice were housed in social isolation (one mouse per cage, all cage conditions otherwise identical to those of group-housed animals) for at least 2 weeks, and then tested again in the resident-intruder assay with an unfamiliar male intruder. Two Nlgn3 single-housed het animals were excluded from analysis due to corruption of behavior videos by acquisition software. One Chd8 GH wt video was excluded due to uncharacteristic aggression by the BalbC intruder mouse.

Videos of social interactions were scored on a frame-by-frame basis for mounting, attack, and close investigation behavior using MARS; select videos were also manually annotated for these three behaviors to confirm the accuracy of MARS classifier output. Manual annotation of this dataset was performed blinded to animal genotype.

## Statistical analysis of autism-associated mutation lines

Testing for effect of mouse cohort (heterozygous mutant or strain vs. wild-type controls) on social behavior expression was performed using a two-tailed t-test. Because the same animals were tested in both group-housed (GH) and singly housed (SH) conditions, testing for effect of housing on social behavior expression was performed using a paired two-tailed t-test. The values tested (total time spent engaging in behavior and average duration of behavior bouts) are approximately Gaussian, justifying the use of the t-test in this analysis (*Table 2*).

## Mouse pose annotation

Part keypoint annotations are common in computer vision and are included in datasets such as Microsoft COCO (*Lin et al., 2014*), MPII human pose (*Andriluka et al., 2014*), and CUB-200-2011 (*Wah et al., 2011*); they also form the basis of markerless pose estimation systems such as DeepLabCut (*Mathis et al., 2018*), LEAP (*Pereira et al., 2019*), and DeepPoseKit (*Graving et al., 2019*). For MARS, we defined 9 anatomical keypoints in the top-view video (the nose, ears, base of neck, hips, and tail base, midpoint, and endpoint) and 13 keypoints in the front-view video (top-view keypoints plus the four paws). The tail mid- and endpoint annotations were subsequently discarded for training of MARS, leaving 7 keypoints in the top view and 11 in the front view (as in *Figure 2A and B*).

To create a dataset of video frames for labeling, we sampled 64 videos from several years of experimental projects in the Anderson lab, collected by multiple lab members (these videos were distinct from the videos used for behavior classification). While all videos were acquired in our standardized hardware setup, we observed some variability in lighting and camera contrast across the dataset; examples are shown in *Figure 1B*. We extracted a set of 15,000 individual frames each from the top- and front-view cameras, giving a total of 2,700,000 individual keypoint annotations (15,000 frames × (7 top-view + 11 front-view keypoints per mouse) × 2 mice × 5 annotators). 5000 of the extracted frames included resident mice with a fiberoptic cable, cannula, or head-mounted microendoscope with cable.

We used the crowdsourcing platform Amazon Mechanical Turk (AMT) to obtain manual annotations of pose keypoints. AMT workers were provided with written instructions and illustrated examples of each keypoint, and instructed to infer the location of occluded keypoints. Frames were assigned randomly to AMT workers and were provided as images (i.e., with no temporal information). Each worker was allowed to annotate as many frames as desired until the labeling job was completed. Python scripts for creation of AMT labeling jobs and postprocessing of labeled data are included in the MARS_Developer repository.

To compensate for annotation noise, each keypoint was annotated by five AMT workers and a 'ground truth' location for that keypoint was defined as the median across annotators (see next section) (*Figure 2C and D*). The median was computed separately in the x and y dimensions. Annotations of individual workers were also postprocessed to correct for common mistakes, such as confusing the left and right sides of the animals. Another common worker error was to mistake the top of the head-mounted microendoscope for the resident animal's nose; we visually screened for these errors and corrected them manually.

## Consolidating data across annotators

We evaluated four approaches for consolidating annotator estimates of keypoint locations: the mean, median, geometric median (*Vardi and Zhang, 2000*), and medoid, using simulated data. We simulated a 10,000-frame dataset with n = 5 simulated clicks per frame, in which clicks were scattered around a ground-truth location with either normal or standard Cauchy-distributed noise; the latter was selected for its heavy tail compared to the normal distribution. Across 100 instantiations of this simulation, the mean error between estimated keypoint location and ground truth was as follows (mean ± STD across 100 simulations):

| Noise type | Mean | Median | Geometric median | Medoid |
|---|---|---|---|---|
| Normal | 0.561 ± 0.00313 | 0.671 ± 0.00376 | 0.646 ± 0.00367 | 0.773 ± 0.00444 |
| Cauchy | 20.6 ± 42.4 | 1.12 ± 0.011 | 1.13 ± 0.0121 | 1.43 ± 0.0125 |

While averaging annotator clicks works well for normally distributed data, it fails when click locations have a heavy tail, which can occur when there is variation in annotator skill or level of focus. We selected the median for use in this study as it performs comparably to the geometric median while being simpler to implement and faster to compute.

## Bounding box annotation

For both top- and front-view video, we estimated a bounding box by finding the minimal rectangle that contained all 7 (top) or 11 (front) pose keypoints. (For better accuracy in the detection and pose estimation, we discarded the middle and end keypoints of the tail.) We then padded this minimal rectangle by a constant factor to prevent cutoff of body parts at the rectangle border.

## Mouse behavior annotation

We created an approximately 14 hr dataset of behavior videos, compiled across recent experiments performed by a member of the Anderson lab; this same lab member ('human 1' from the multi-annotator dataset) annotated all videos on a frame-by-frame basis. The videos in this dataset were largely distinct from the videos sampled to create the 15,000-frame pose dataset. Annotators were provided with simultaneous top- and front-view video of interacting mice, and scored every video frame for close investigation, attack, and mounting, using the criteria described below. In some videos, additional behaviors were also annotated; when this occurred, these behaviors were assigned to one of close investigation, attack, mounting, or 'other' for the purpose of training classifiers. Definitions of these additional behaviors are listed underneath the behavior to which they were assigned. All behavior definitions reflect an Anderson lab consensus, although as evidenced by our multi-annotator comparison, even such a consensus does not prevent variability in annotation style across individuals. Annotation was performed either in BENTO or using a previously developed custom MATLAB interface (*Dollar, 2016*).

Close investigation: Resident (black) mouse is in close contact with the intruder (white) and is actively sniffing the intruder anywhere on its body or tail. Active sniffing can usually be distinguished from passive orienting behavior by head bobbing/movements of the resident's nose.

Other behaviors converted to the 'close investigation' label:

- Sniff face: Resident investigation of the intruder's face (typically eyes and snout).
- Sniff genitals: Resident investigation of the intruder's anogenital region, often occurs by shoving of the resident's snout underneath the intruder's tail.
- Sniff body: Resident investigation of the intruder's body, anywhere away from the face or genital regions.
- Attempted attack: This behavior was only annotated in a small subset of videos and was grouped with investigation upon visual investigation of annotated bouts and comparison to annotations in other videos. Intruder is in a defensive posture (standing on hind legs, often facing the resident) to protect itself from attack, and resident is close to the intruder, either circling or rearing with front paws out towards/on the intruder, accompanied by investigation. Typically follows or is interspersed with bouts of actual attack; however, the behavior itself more closely resembles investigation.
- Attempted mount (or attempted dom mount): This behavior was only annotated in a small subset of videos and was grouped with investigation upon visual investigation of annotated

bouts and comparison to annotations in other videos. Resident attempts to climb onto or mount another animal, often accompanied by investigation. Palpitations with forepaws and pelvic thrusts may be present, but the resident is not aligned with the body of the intruder mouse or the intruder mouse may be unreceptive and still moving.

Attack: High-intensity behavior in which the resident is biting or tussling with the intruder, including periods between bouts of biting/tussling during which the intruder is jumping or running away and the resident is in close pursuit. Pauses during which resident/intruder are facing each other (typically while rearing) but not actively interacting should not be included.

Mount: Copulatory behavior in which the resident is hunched over the intruder, typically from the rear, and grasping the sides of the intruder using forelimbs (easier to see on the front camera). Early-stage copulation is accompanied by rapid pelvic thrusting, while later-stage copulation (sometimes annotated separately as intromission) has a slower rate of pelvic thrusting with some pausing: for the purpose of this analysis, both behaviors should be counted as mounting; however, periods where the resident is climbing on the intruder but not attempting to grasp the intruder or initiate thrusting should not.

Other behaviors converted to the 'mount' label:

- Intromission: Late-stage copulatory behavior that occurs after mounting, with a slower rate of pelvic thrusting. Occasional pausing between bouts of thrusting are still counted as intromission.
- Dom mount (or male-directed mounting): This behavior was only annotated in a small subset of videos. Visually similar behavior to mounting; however, typically directed towards a male intruder. The primary feature that distinguishes this behavior from mounting is an absence of ultrasonic vocalizations; bouts are also typically much shorter in duration and terminated by the intruder attempting to evade the resident.

Other: Behaviors that were annotated in some videos but not included in any of the above categories.

- Approach: Resident orients and walks toward a typically stationary intruder, typically followed by periods of close investigation. Approach does not include more high-intensity behavior during which the intruder is attempting to evade the resident, which is instead classified as chase.
- Chase: Resident is closely following the intruder around the home cage, while the intruder attempts to evade the resident. Typically interspersed with attempted mount or close investigation. In aggressive encounters, short periods of high-intensity chasing between periods of attack are still labeled as attack (not chase), while longer periods of chasing that do not include further attempts to attack are labeled as chasing.
- Grooming: Usually in a sitting position, the mouse will lick its fur, groom with the forepaws, or scratch with any limb.

## Design of behavior training, validation, and test sets

Videos were randomly assigned to train/validation/test sets by resident mouse identity, that is, all videos of a given resident mouse were assigned to the same dataset. This practice is preferable to random assignment by video frame because the latter can lead to temporally adjacent (and hence highly correlated) frames being distributed into training and test sets, causing severe overestimation of classifier accuracy. (For example, in fivefold cross-validation with train/test set assignments randomized by frame, 96% of test-set frames will have an immediately neighboring frame in the training set.) In contrast, randomization by animal identity ensures that we do not overestimate the accuracy of MARS classifiers and best reflects the expected accuracy end-users can expect when recording under comparable conditions, as the videos in the test set are from mice that MARS has never encountered before.

Note that because data were randomized by animal identity, relative frequencies of behaviors show some variation between training, validation, and test sets. Furthermore, because some videos (most often miniscope experiments) were annotated in a more granular manner than others, some sets (e.g., test set 1) are dominated by attack/mount/sniff annotations, while other sets include more annotations from other behaviors.

## Behavior annotation by multiple individuals

For our analysis of inter-annotator variability in behavior scoring, we provided a group of graduate students, postdocs, and technicians in the Anderson lab with the descriptions of close investigation, mounting, and attack given above, and instructed them to score a set of 10 resident-intruder videos, all taken from unoperated mice. Annotators were given front- and top-view video of social interactions, and scored behavior using either BENTO or the Caltech Behavior Annotator (*Dollar, 2016*), both of which support simultaneous display of front- and top-view video and frame-by-frame browsing and scoring. All but one annotator (human 4) had previous experience scoring mouse behavior videos; human 4 had previous experience scoring similar social behaviors in flies.

When comparing human annotations to 'annotator median,' each annotator was compared to the median of the remaining annotators. When taking the median of six annotators, a frame was considered positive for a given behavior if at least three out of six annotators labeled it as positive.

## The MARS pipeline

### Overview

MARS processes videos in three steps. First, videos are fed frame-by-frame into detection and pose estimation neural networks (details in following sections). Frames are loaded into a queue and passed through a set of six functions: detection preprocessing, detection, detection postprocessing, pose preprocessing, pose estimation, and pose postprocessing; output of each function is passed into an input queue for the next. MARS uses multithreading to allow each stage in the detection and pose estimation pipeline to run independently on its queued frames, reducing processing time. Pose estimates and bounding boxes are saved every 2000 frames into a json file.

Second, following pose estimation, MARS extracts a set of features from estimated poses and the original tracked video (for pixel-change features; details in following sections). The MARS interface allows users to extract several versions of features; however, this paper focuses only on the 'top' version of features as it requires only top-view video input. Other tested feature sets, which combined the 270 MARS features with additional features extracted from front-view video, showed little improvement in classifier performance; these feature sets are still provided as options in the MARS user interface for potential future applications. The 270 MARS features are extracted and saved as .npz and .mat files (for use with MARS and BENTO, respectively). MARS next applies temporal windowing to these features (see following sections) and saves them as separate .npz and .mat files with a '_wnd' suffix.

Third, following feature extraction, MARS loads the windowed version of features and uses these as input to a set of behavior classifiers (details in following sections). The output of behavior classifiers is saved as a .txt file using the Caltech Behavior Annotator format. In addition, MARS generates an Excel file that can be used to load video, annotations, and pose estimates into BENTO.

### Mouse detection using the multi-scale convolutional MultiBox detector

We used the multi-scale convolutional MultiBox (MSC-MultiBox) (*Erhan et al., 2014*; *Szegedy et al., 2014*) approach to train a pair of deep neural networks to detect the black and white mice using our 15,000 frame bounding box annotation dataset. Specifically, we used the Inception-Resnet-v2 architecture (*Szegedy et al., 2017*) with ImageNet pretrained weights, trained using a previously published implementation of MSC-MultiBox for TensorFlow (https://github.com/gvanhorn38/multibox).

Briefly, MSC-MultiBox computes a short list of up to K possible object detections proposal (bounding boxes) and associated confidence scores denoting the likelihood of that box containing a target object, in this case the black or white mouse. During training, MSC-MultiBox seeks to optimize location and maximize confidence scores of predicted bounding boxes that best match the ground truth, while minimizing confidence scores of predicted bounding boxes that do not match the ground truth. Bounding box location is encoded as the coordinates of the box's upper-left and lower-right corners, normalized with respect to the image dimensions; confidence scores are scaled between 0 (lowest) and 1 (highest). Once we have the predicted bounding box proposals and confidence score, we used non-maximum suppression (NMS) to select the bounding box proposal that best matches with the ground truth.

During training, we augmented data with random color variation and left/right flips. We used a learning rate of 0.01, decayed exponentially by 0.94 every four epochs, with an RMSProp optimizer

with momentum and decay both set to 0.99 and batch size of 4. Video frames are resized to 299 × 299 during both training and inference. The model was trained on an 8-core Intel i7-6700K CPU with 32 GB RAM and an 8 GB GTX 1080 GPU. All parameters used during training are published online in the detection model config_train.yaml file published in the MARS_Developer repository at https://github.com/neuroecology/MARS_Developer.

### Detector evaluation

Detectors were trained on 12,750 frames from our pose annotation dataset, validated using 750 frames, and evaluated on the remaining 1500 held-out frames, which were randomly sampled from the dataset.

Model evaluation was performed using the COCO API (*Lin et al., 2014*). We evaluated performance of the MSC-MultiBox detector by computing the IoU between estimated and human-annotated bounding boxes, defined as the area of the intersection of the human-annotated and estimated bounding boxes, divided by the area of their union. PR curves were plotted based on the fraction of MSC-MultiBox detected bounding boxes with an IoU > X, for X in (0.5, 0.75, 0.85, 0.9, 0.95).

### Pose estimation

Following the detection step (above), we use the stacked hourglass network architecture (*Newell et al., 2016*) to estimate the pose of each mouse in terms of a set of anatomical keypoints (7 keypoints in top-view videos and 11 keypoints in front-view videos) on our 15,000-frame keypoint annotation dataset. We selected the stacked hourglass architecture for its high performance on human pose estimation tasks. The network's repeated 'hourglass' modules shrink an input image to a low resolution, then up-samples it while combining it with features passed via skip connections; representations from multiple scaled versions of the image are thus combined to infer keypoint location. We find that the stacked hourglass network is robust to partial occlusion of the animals using the visible portion of a mouse's body to infer the location of parts that are hidden.

To construct the input to the stacked hourglass network, MARS crops each video frame to the bounding box of a given mouse plus an extra 65% width and height, pad the resulting image with zero-value pixels to make it square, and resize to 256 × 256 pixels. Because the stacked hourglass network converts an input image to a heatmap predicting the probability of a target keypoint being present at each pixel, during network training we constructed training heatmaps as 2D Gaussians with standard deviation of 1 px centered on each annotator-provided keypoint. During inference on user data, MARS takes the maximum value of the generated heatmap to be the keypoint's location. We trained a separate model for pose estimation in front- vs. top-view videos, but for each view the same model was used for both the black and the white mice.

To improve generalizability of MARS pose estimation, we used several data augmentation manipulations to expand the effective size of our training set, including random blurring (p=0.15, Gaussian blur with standard deviation of 1 or 2 pixels), additive Gaussian noise (pixelwise across image with p=0.15), brightness/contrast/gamma distortion (p=0.5), and jpeg artifacts (p=0.15), random rotation (p=1, angle uniform between 0 and 180), and random padding of the bounding box and random horizontal and vertical flipping (p=0.5 each).

During training, we used an initial learning rate of 0.00025, fixed learning rate decay by a factor of 0.2 every 33 epochs to a minimum learning rate of 0.0001, and batch size of 8 (all parameters were based on the original stacked hourglass paper; *Newell et al., 2016*). For optimization, we use the RMSProp optimizer with momentum of 0, decay of 0.9. The network was trained using TensorFlow on an 8-core Intel Xeon CPU, with 24 Gb RAM and a 12 GB Titan XP GPU. All parameters used during training are published online in pose and detection model training conFigure files, filename config_train.yaml located in the MARS_Developer repository at https://github.com/neuroecology/MARS_Developer.

### Pose evaluation

Each pose network was trained on 13,500 video frames from our pose annotation dataset and evaluated on the remaining 1500 held-out frames, which were randomly sampled from the full dataset. The same held-out frames were used for both detection and pose estimation steps.

We evaluated the accuracy of the MARS pose estimator by computing the 'percent correct keypoints' metric from COCO (*Lin et al., 2014*), defined as the fraction of predicted keypoints on test frames that fell within a radius X of 'ground truth.' Ground truth for this purpose was defined as the median of human annotations of keypoint location, computed along x and y axes separately.

To summarize these curves, we used the OKS metric introduced by *Ruggero Ronchi and Perona, 2017*. Briefly, OKS is a measure of pose accuracy that normalizes errors by the estimated variance of human annotators. Specifically, given a keypoint with ground-truth location X and estimated location X, the OKS for a given body part is defined as

$$OKS = e^{\frac{-(\hat{X}-X)^2}{2\sigma^2 k^2}}$$

Here, $k^2$ is the size of the instance (the animal) in pixels, and $\sigma^2$ is the variance of human annotators for that body part. Thus, an error of Z pixels is penalized more heavily for body parts where human variance is low (such as the nose), and more leniently for body parts where the ground truth itself is more unclear and human variance is higher (such as the sides of the body). We computed $\sigma$ for each body part from our 15,000-frame dataset, in which each frame was annotated by five individuals. This yielded the following values of $\sigma$: for the top-view camera, 'nose tip': 0.039; 'right ear': 0.045; 'left ear': 0.045; 'neck': 0.042; 'right-side body': 0.067; 'left-side body': 0.067; 'tail base': 0.044; 'middle tail': 0.067; 'end tail': 0.084. For the front-view camera, 'nose tip': 0.087; 'right ear': 0.087; 'left ear': 0.087; 'neck': 0.093; 'right-side body': 0.125; 'left-side body': 0.125; 'tail base': 0.086; 'middle tail': 0.108; 'end tail': 0.145; 'right front paw': 0.125; 'left front paw': 0.125' 'right rear paw': 0.125; 'left rear paw': 0.125. We also computed OKS values assuming a fixed $\sigma = 0.025$ for all body parts, as reported in SLEAP (*Pereira et al., 2020b*).

OKS values are typically summarized in terms of the mAP and mAR (*Pishchulin et al., 2016*), where precision is true positives/(true positives + false positives), and recall is true positives/(true positives + false negatives). For pose estimation, a true positive occurs when a keypoint is detected falls within some 'permissible radius' R of the ground truth.

To distinguish False positives from false negatives, we take advantage of the fact that MARS returns a confidence S for each keypoint, reflecting the model's certainty that a keypoint was indeed at the provided location. (MARS's pose model will return keypoint locations regardless of confidence; however, low confidence is often a good indicator that those locations will be less accurate.) We will therefore call a keypoint a false positive if confidence is above some threshold C but location is far from ground truth, and a false negative otherwise. Because there is always a ground-truth keypoint location (even when occluded), there is no true negative category.

Given some permissible radius R, we can thus plot precision-recall curves as one would for a classifier, by plotting precision vs. recall as we vary the confidence threshold C from 0 to 1. We summarize this plot by taking the area under the P-R curve, a value called the average precision (AP). We also report the fraction of true positive detections if any confidence score is accepted—called the average recall (AR).

The last value to set is our permissible radius R: how close does a predicted keypoint have to be to ground truth to be considered correct, and with what units? For units, we will use our previously defined OKS, which ranges from 0 (poor) to 1 (perfect). For choice of R, the accepted approach in machine learning (*Pishchulin et al., 2016*; *Lin et al., 2014*) is to compute the AP and AR for each of R = [0.5,0.55,0.6,0.65,0.7,0.75,0.8,0.85,0.9,0.95], and then to take the mean value of AP and AR across these 10 values, thus giving the mAP and mAR.

## Pose features

Building on our previous work (*Hong et al., 2015*), we designed a set of 270 spatiotemporal features extracted from the poses of interacting mice, to serve as input to supervised behavior classifiers. MARS's features can be broadly grouped into locomotion, position, appearance, social, and image-based categories. *Position features* describe the position of an animal in relation to landmarks in the environment, such as the distance to the wall of the arena. *Appearance-based features* describe the pose of the animal in a single frame, such as the orientation of the head and body or the area of an ellipse fit to the animal's pose. *Locomotion features* describe the movement of a single animal, such as speed or change of orientation of the animal. *Image-based* features describe the change of pixel

intensity between movie frames. Finally, *social features* describe the position or motion of one animal relative to the other, such as inter-animal distance or difference of orientation between the animals. A full list of extracted features and their definitions can be found in *Table 3*. Most features are computed for both the resident and intruder mouse; however, a subset of features are identical for the two animals and are computed only for the resident, as indicated in the table.

Features are extracted for each frame of a movie, then each feature is smoothed by taking a moving average over three frames. Next, for each feature we compute the mean, standard deviation, minimum, and maximum value of that feature in windows of ±33, ±167, and ±333 ms relative to the current frame, as in *Kabra et al., 2013*; this addition allows MARS to capture how each feature is evolving over time. We thus obtain 12 additional 'windowed' features for each original feature; we use 11 of these (omitting the mean of the given feature over ±1 frame) plus the original feature as input to our behavior classifiers, giving a total of 3144 features. For classification of videos with different framerates from the training set, pose trajectories are resampled to match the framerate of the classifier training data.

In addition to their use for behavior classification, the pose features extracted by MARS can be loaded and visualized within BENTO, allowing users to create custom annotations by applying thresholds to any combination of features. MARS features include many measurements that are commonly used in behavioral studies, such as animal velocity and distance to arena walls (*Table 3*).

### Behavior classifiers

From our full 14 hr video dataset, we randomly selected a training set of 6.95 hr of video annotated on a frame-by-frame basis by a single individual (human #1 in *Figure 5*) for close investigation, mounting, and attack behavior. From these annotated videos, for each behavior we constructed a training set (**X, y**) where $X_i$ corresponds to the 3144 windowed MARS pose features on frame $i$, and $y_i$ is a binary label indicating the presence or absence of the behavior of interest on frame $i$. We evaluated performance of and performed parameter exploration using a held-out validation set of videos. A common form of error in many of our tested classifiers was to have sequences (1–3 frames) of false negative or false positives that were shorter than the typical behavior annotation bout. To correct these short error bouts, we introduced a postprocessing stage following frame-wise classification, in which the classifier prediction is smoothed using an HMM followed by a three-frame moving average.

In preliminary exploration, we found that high precision and recall values for individual binary behavior classifiers were achieved by gradient boosting using the XGBoost algorithm (*Chen and Guestrin, 2016*); we therefore used this algorithm for the three classifiers presented in this paper. Custom Python code to train novel behavior classifiers is included with the MARS_Developer software. Classifier hyperparameters may be set by the user, otherwise MARS will provide default values.

Each trained classifier produces a predicted probability that the behavior occurred, as well as a binarized output created by thresholding that probability value. Following predictions by individual classifiers, MARS combines all classifier outputs to produce a single, multi-class label for each frame of a behavior video. To do so, we select on each frame the behavior label that has the highest predicted probability of occurring; if no behavior has a predicted probability of >0.5, then the frame is labeled as 'other' (no behavior occurring). The advantage of this approach over training multi-class XGBoost is that it allows our ensemble of classifiers to be more easily expanded in the future to include additional behaviors of interest because it does not require the original training set to be fully re-annotated for the new behavior.

### Classifier evaluation

Accuracy of MARS behavior classifiers was estimated in terms of classifier precision and recall, where precision = (number of true positive frames)/(number of true positive and false positive frames), and recall = (number of true positive frames)/(number of true positive and false negative frames). Precision and recall scores were estimated for the set of trained binary classifiers on a held-out test set of videos not seen during classifier training. PR curves were created for each behavior classifier by calculating classifier precision and recall values as the decision threshold (the threshold for classifying a frame as positive for a behavior) is varied from 0 to 1.

## Acknowledgements

We are grateful to Grant Van Horn for providing the original TensorFlow implementation of the MSC-MultiBox detection library, Matteo Ronchi for his pose error diagnosis code, and Mark Zylka for providing the Cul3 and Chd8 mouse lines. Research reported in this publication was supported by the National Institute of Mental Health of the National Institutes of Health under Award Number R01MH123612 and 5R01MH070053 (DJA), K99MH108734 (MZ) and K99MH117264 (AK), and by the Human Frontier Science Program (TK), the Helen Hay Whitney Foundation (AK), the Simons Foundation Autism Research Initiative (DJA), the Gordon and Betty Moore Foundation (PP), and a gift from Liying Huang and Charles Trimble (to PP). The content is solely the responsibility of the authors and does not necessarily represent the official views of the National Institutes of Health.

## Additional information

### Funding

| Funder | Grant reference number | Author |
|---|---|---|
| National Institute of Mental Health | K99MH117264 | Ann Kennedy |
| National Institute of Mental Health | K99MH108734 | Moriel Zelikowsky |
| Simons Foundation Autism Research Initiative | 401141 | David J Anderson |
| Helen Hay Whitney Foundation | Postdoctoral Fellowship | Ann Kennedy |
| Human Frontier Science Program | Long-term Fellowship | Tomomi Karigo |
| National Institute of Mental Health | R01MH123612 | David J Anderson |
| Gordon and Betty Moore Foundation | | Pietro Perona |
| Liying Huang and Charles Trimble | | Pietro Perona |
| Simons Foundation | Simons Collaboration on the Global Brain Foundation 542947 | David J Anderson Pietro Perona |

The funders had no role in study design, data collection and interpretation, or the decision to submit the work for publication.

### Author contributions

Cristina Segalin, Conceptualization, Data curation, Investigation, Methodology, Software, Supervision, Visualization, Writing – original draft; Jalani Williams, Investigation, Software, Visualization; Tomomi Karigo, May Hui, Data curation, Investigation; Moriel Zelikowsky, Data curation, Supervision; Jennifer J Sun, Investigation; Pietro Perona, Conceptualization, Funding acquisition, Methodology, Project administration, Writing – original draft; David J Anderson, Conceptualization, Funding acquisition, Project administration, Writing – original draft; Ann Kennedy, Conceptualization, Data curation, Investigation, Project administration, Software, Visualization, Writing – original draft

### Author ORCIDs

Cristina Segalin http://orcid.org/0000-0001-7219-7074
May Hui http://orcid.org/0000-0002-6231-7383
Moriel Zelikowsky http://orcid.org/0000-0002-0465-9027
Jennifer J Sun http://orcid.org/0000-0002-0906-6589
David J Anderson http://orcid.org/0000-0001-6175-3872
Ann Kennedy http://orcid.org/0000-0002-3782-0518

### Ethics

All experimental procedures involving the use of live animals or their tissues were performed in accordance with the recommendations in the Guide for the Care and use of Laboratory animals of the National Institutes of Health. All of the animals were handled according to the Institutional Animal Care and Use Committee (IACUC) protocols (IA18-1552); the protocol was approved by the Institutional Biosafety Committee at the California Institute of Technology (Caltech).

### Decision letter and Author response

Decision letter https://doi.org/10.7554/eLife.63720.sa1
Author response https://doi.org/10.7554/eLife.63720.sa2

## Additional files

### Supplementary files

• Transparent reporting form

### Data availability

All data used to train and test MARS is hosted by the Caltech library at data.caltech.edu. MARS code is publicly available on Github: - Core end-user version of MARS: http://github.com/neuroethology/MARS - Code for training new MARS models: http://github.com/neuroethology/MARS_Developer - Bento interface for browsing data: http://github.com/neuroethology/bentoMAT.

The following dataset was generated:

| Author(s) | Year | Dataset title | Dataset URL | Database and Identifier |
|---|---|---|---|---|
| Segalin C, Williams J, Karigo T, Hui M, Zelikowski M, Sun JJ, Perona P, Anderson DJ, Kennedy A | 2021 | The Mouse Action Recognition System (MARS): pose annotation data (Version 1.0) [Data set] | https://doi.org/10.22002/D1.2011 | CaltechDATA, 10.22002/D1.2011 |
| Segalin C, Williams J, Karigo T, Hui M, Zelikowski M, Sun JJ, Perona P, Anderson DJ, Kennedy A | 2021 | The Mouse Action Recognition System (MARS): behavior annotation data (Version 1.0) [Data set] | https://doi.org/10.22002/D1.2012 | CaltechDATA, 10.22002/D1.2012 |
| Segalin C, Williams J, Karigo T, Hui M, Zelikowski M, Sun JJ, Perona P, Anderson DJ, Kennedy A | 2021 | The Mouse Action Recognition System (MARS): multi-worker behavior annotation data (Version 1.0) [Data set] | https://doi.org/10.22002/D1.2121 | CaltechDATA, 10.22002/D1.2121 |

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
