## [Decision Letter]

**Acceptance summary:**

Segalin and colleagues present a pair of open-source software tools – MARS and BENTO – for automatic pose detection, social behavior detection, and interactive neural/behavior visualization in mice. MARS builds on previous tools for social behavior annotation, but now eliminating the need for two cameras and for a depth signal, incorporating deep learning, and building in robustness to neural implants. BENTO further extends this work by adding a suite of tools for annotating video frames, visualizing neural activity, and performing simple operations on the neural activity data such as event-triggered averaging. Importantly, Segalin and colleagues also share a large-scale dataset to train this system. Together, these tools will be useful for researchers studying the neural underpinnings of rodent social behavior, in particular with the resident intruder assay.

**Decision letter after peer review:**

Thank you for submitting your article "The Mouse Action Recognition System (MARS): a software pipeline for automated analysis of social behaviors in mice" for consideration by *eLife*. Your article has been reviewed by 3 peer reviewers, and the evaluation has been overseen by a Reviewing Editor and Kate Wassum as the Senior Editor. The following individual involved in review of your submission has agreed to reveal their identity: Asaf Gal (Reviewer #3).

The reviewers have discussed the reviews with one another and the Reviewing Editor has drafted this decision to help you prepare a revised submission.

Summary:

Segalin and colleagues present a pair of open-source software tools – MARS and BENTO – for automatic pose detection, social behavior detection, and interactive neural/behavior visualization in mice. MARS builds on previous tools for social behavior annotation, but now eliminating the need for two cameras and for a depth signal, incorporating deep learning, and building in robustness to neural implants. BENTO further extends this work by adding a suite of tools for annotating video frames, visualizing neural activity, and performing simple operations on the neural activity data such as event-triggered averaging. Importantly, Segalin and colleagues also share a large-scale dataset to train this system. Together, these tools will be useful for researchers studying the neural underpinnings of rodent social behavior, in particular with the resident intruder assay.

The reviewers were generally enthusiastic about this submission, but would like to see several revisions before recommending for acceptance in *eLife*. An additional point, however, was the reviewers thought that this article was more suitable as a "Tools and Resources" article (see guidelines here: https://reviewer.elifesciences.org/author-guide/types ), so we ask the authors to make their revisions with this classification in mind.

Essential revisions:

1) The authors promise at several points that features will become available in the future, including:

(1) A Python implementation of BENTO (MARS is already implemented in Python, whereas BENTO is currently implemented in Matlab);

(2) The ability the ability to detect behaviors in pairs of mice with the same coat color;

(3) The ability to train MARS on one's own behavior annotations.

While features 1 and 2 can, of course, wait – software takes time to develop – the absence of feature 3 is a little more confusing. BENTO appears to include an interface for annotating frames. Is it not possible for MARS to read these annotations and to include a framework in which a new classifier is trained? This is important because the repertoire of behaviors captured by the MARS classifier is limited. The previous rodent social behavior detection papers cited by the authors (e.g. refs 29,30 and 33) include a much richer menu of behavior labels (for example SimBA [ref 33] includes "attack, pursuit, lateral threat, anogenital sniffing, allogrooming normal, allogrooming vigorous, mounting, scramble, flee, and upright submissive"). Moreover, the authors have richer annotations available within their own training data. In the Methods, the authors note that many of the original video annotations used for this study actually did include a much higher resolution of labels, but that these labels were collapsed for training MARS. For example, the frames labeled "close investigation" are actually the union of five different labeled categories. "Sniff face, Sniff genitals, Sniff body, Attempted attack, Attempted mount. Why were these combined, given that I would expect them to have very different neurobiological correlates. For example "attempted attack" would appear to have more in common with "Attack" than with "Sniff body".

2) The authors argue at several points that supervised classification can benefit the neuroscience community by creating a common definition of social behaviors would be interoperable between labs, that could, for example "facilitate comparison of behavioral results between labs, fostering large‐scale screening of social behaviors within a common analysis platform." This paper would be stronger if the authors could spell out a formula for finding consensus between annotators; given these principles, perhaps MARS could be trained to reflect this consensus. Perhaps trained MARS models could contain one or more tuning parameters, so that every annotator could be captured by some value(s) of the parameters, thus providing a unified framework while accommodating individual variation in annotation habits?

3) Multiple datasets are provided, one particularly novel one is the 10 x 10min videos of resident intruder assays, annotated by 8 individuals. It would be spectacular if those videos could be annotated a second time by the same individuals. Mainly, to see if individuals are consistent with themselves (stable style). This would nicely complement the dataset. In the description of the datasets, the relationships are sometimes not clear, e.g. is the person that annotated the large-scale dataset of ~14h also one of the 8 individuals and if yes, then which one?

4) The reviewers wanted to make sure that the 14 hour dataset will be shared (this was not clear from the manuscript). Moreover, when the reviewers attempted to download the dataset it was corrupted (this was replicated by two reviewers and the reviewing editor). In addition, the reviewers wanted to make sure that the data contains all the appropriate meta-data (e.g., annotation images/videos, which genotype the mice have, when the data was recorded, camera type, etc. )

5) The analyzed datasets and training data were all collected in the same lab, using the same standard setup, under very similar conditions that do not capture the variability expected across the many labs that use this assay. In order to be useful to other users, the conditions at which the method will work must be explicitly discussed. This is especially concerning, as the version of MARS presented in the paper does not allow users to define their own pipelines or to fine-tune the supplied one with new data. Furthermore, the feature list of the behavioral classifiers contains features that are very setup specific (e.g., distance to arena wall, absolute coordinates etc.), or organism-line specific (e.g. mouse size/area). If the authors indeed aim at creating a standardized behavior classification pipeline, which can be compared across labs, it is essential to show/discuss how the method gives consistent results across at least some of the different experimental variants of this assay.

6) Although the paper describes the algorithm well, if was hard for the reviewers to judge the actual usability and quality of the tool without access to it. The software should be made available to the reviewers (with documentation).

7) While a direct comparison to other social tracking methods (e.g., maDLC, SLEAP, others) is not necessary here, it is important to have a more comprehensive error analysis of the tracking (while the animals are in the bounding box, minimally).

8) Please ensure full statistical reporting in the main manuscript (e.g., t, F values, degrees of freedom, p value, etc.).

---

## [Author Response]

Essential revisions:1) The authors promise at several points that features will become available in the future, including:(1) A Python implementation of BENTO (MARS is already implemented in Python, whereas BENTO is currently implemented in Matlab);(2) The ability the ability to detect behaviors in pairs of mice with the same coat color;(3) The ability to train MARS on one's own behavior annotations.While features 1 and 2 can, of course, wait – software takes time to develop – the absence of feature 3 is a little more confusing. BENTO appears to include an interface for annotating frames. Is it not possible for MARS to read these annotations and to include a framework in which a new classifier is trained? This is important because the repertoire of behaviors captured by the MARS classifier is limited. The previous rodent social behavior detection papers cited by the authors (e.g. refs 29,30 and 33) include a much richer menu of behavior labels (for example SimBA [ref 33] includes "attack, pursuit, lateral threat, anogenital sniffing, allogrooming normal, allogrooming vigorous, mounting, scramble, flee, and upright submissive"). Moreover, the authors have richer annotations available within their own training data. In the Methods, the authors note that many of the original video annotations used for this study actually did include a much higher resolution of labels, but that these labels were collapsed for training MARS. For example, the frames labeled "close investigation" are actually the union of five different labeled categories. "Sniff face, Sniff genitals, Sniff body, Attempted attack, Attempted mount. Why were these combined, given that I would expect them to have very different neurobiological correlates. For example "attempted attack" would appear to have more in common with "Attack" than with "Sniff body".

We thank the reviewers for this important feedback, and have attempted to address it point by point below.

First, regarding the repertoire of behaviors captured by MARS: we chose to focus on attack, mounting, and investigation in this manuscript because (a) these were the three behaviors most consistently annotated in our training and test sets, and (b) these were the three behaviors for which we could directly compare MARS performance to inter-human annotation variability. The other behaviors indicated in the Methods were present only in a subset of videos- typically in the minority of videos with accompanying microendoscopic imaging.

Nonetheless, to show that MARS can detect other behaviors, we have added text (lines 369-375) and a figure (Figure 6- figure supplement 3) to our resubmission showing performance of classifiers trained for intromission, sniff face, and sniff genital, the three behaviors with cleanest annotations in our dataset.

Furthermore, we agree with the reviewers that support for retraining MARS would increase the impact of this paper. We have therefore introduced a new Github repository, MARS_Developer, that allows users to train detection models, pose models, and behavior classifiers. The repository includes a detailed MARS_tutorial jupyter notebook that walks users through the steps in training and explains the metrics used for model evaluation in depth. This addition allows MARS to be adapted to function in diverse settings, and lets users train their own behavior classifiers using annotations produced in BENTO. We have also included a button in the BENTO interface to send annotations directly to MARS (lines 385-386 of text).

MARS_Developer is still in active development for example, we plan to integrate support for our self-supervised behavior learning tool, TREBA (Sun et al. CVPR 2021), however is currently capable of reproducing the original version of MARS from end-to-end, and supports the training of new classifiers for social behavior data.

Finally, regarding “attempted attack” and “attempted mount”: these were very loosely defined annotations made during exploratory analysis of imaging data; because they are rare, we could not train supervised classifiers to detect them with any reasonable accuracy (nor do we have a great sense of how reliable human annotations for these behaviors would be.) We assigned these behaviors to “investigation” based on close visual inspection, in which we judged how these behaviors would have been otherwise annotated- frames annotated “attempted” mount/attack can best be described as investigation that is somewhat more vigorous, with an apparent “intent” to attack or mount the other mouse.

2) The authors argue at several points that supervised classification can benefit the neuroscience community by creating a common definition of social behaviors would be interoperable between labs, that could, for example "facilitate comparison of behavioral results between labs, fostering large‐scale screening of social behaviors within a common analysis platform." This paper would be stronger if the authors could spell out a formula for finding consensus between annotators; given these principles, perhaps MARS could be trained to reflect this consensus. Perhaps trained MARS models could contain one or more tuning parameters, so that every annotator could be captured by some value(s) of the parameters, thus providing a unified framework while accommodating individual variation in annotation habits?

We agree with the reviewers that finding consensus between annotators or labs is an important goal. We have pursued multiple projects over the past year investigating methods for capturing and explaining inter-annotator variability, the most successful of which was presented at the CV4Animals Workshop at CVPR this year (1). Briefly, we approached this problem as one of “interpretable machine learning”, which is an active area of research (2). Methods in interpretable machine learning fall into two categories: post-hoc interpretation of black-box models, and methods to train inherently interpretable models. Our workshop paper falls into the latter category of creating interpretable models that allow users to visualize differences in classifiers trained to match different individuals’ annotation styles. (We have also worked on the approach reviewers suggest of creating a low-dimensional representation of “annotation style” space, but thus far this has not yielded strong results.) We have included a copy of this manuscript in our resubmission.

But while we are working on the problems the reviewers raise, this is an active area of research and we have not yet arrived at a fully satisfying solution to the problem of reconciling annotator differences. What we have done is (1) provided an easy way to compare pose features and multiple annotations of the same video side-by-side within BENTO (now mentioned in the manuscript, line 386-387), (2) created a BENTO module to visualize models created with our “interpretable” behavior classifiers (this is a part of the workshop paper), and (3) released our 10video x 10min x 8annotator dataset to fuel further research into the area of characterizing inter-annotator differences.

(1) Tjandrasuwita, M., Sun, J. J., Kennedy, A., Chaudhuri, S., and Yue, Y. (2021). Interpreting Expert Annotation Differences in Animal Behavior. arXiv preprint arXiv:2106.06114.

(2) See, eg: Zachary C Lipton (2018) The mythos of model interpretability: In machine learning, the concept of interpretability is both important and slippery; Cynthia Rudin (2019) Stop explaining black box machine learning models for high stakes decisions and use interpretable models instead; Yin Lou et al (2012) Intelligible models for classification and regression.

3) Multiple datasets are provided, one particularly novel one is the 10 x 10min videos of resident intruder assays, annotated by 8 individuals. It would be spectacular if those videos could be annotated a second time by the same individuals. Mainly, to see if individuals are consistent with themselves (stable style). This would nicely complement the dataset. In the description of the datasets, the relationships are sometimes not clear, e.g. is the person that annotated the large-scale dataset of ~14h also one of the 8 individuals and if yes, then which one?

We agree with the reviewers that this is an interesting question! Persuading our 8 annotators to manually re-annotate all 10 videos was a tough sell, as this is a huge amount of work. As a compromise, we selected two videos of the original 10 (one male/male and one male/female), and asked annotators to re-annotate those two; we then evaluated within-annotator vs between-annotator agreement on these videos. Despite the long interval between first and second rounds of annotation (approximately 10 months), we found that annotators typically showed high self consistency: self-self agreement was significantly higher than self-other for both attack and investigation annotations. We have included new text (lines 307-314) and a new ED figure (ED Figure 6) summarizing our analysis of this reannotation dataset, and will include the second round of annotations in our dataset release.

4) The reviewers wanted to make sure that the 14 hour dataset will be shared (this was not clear from the manuscript). Moreover, when the reviewers attempted to download the dataset it was corrupted (this was replicated by two reviewers and the reviewing editor). In addition, the reviewers wanted to make sure that the data contains all the appropriate meta-data (e.g., annotation images/videos, which genotype the mice have, when the data was recorded, camera type, etc. )

We apologize for this issue, we fully intend to make the 14 hour dataset (as well as the 30,000 frame pose annotation dataset, and the 10 videos with annotations by 8 lab members) publicly available. We have relocated all our datasets to data.caltech.edu, and updated the download links in the manuscript. We have also now added mouse genotype, recording dates, and camera information for all videos in this dataset; this information is included in an excel spreadsheet accompanying the videos.

5) The analyzed datasets and training data were all collected in the same lab, using the same standard setup, under very similar conditions that do not capture the variability expected across the many labs that use this assay. In order to be useful to other users, the conditions at which the method will work must be explicitly discussed. This is especially concerning, as the version of MARS presented in the paper does not allow users to define their own pipelines or to fine-tune the supplied one with new data. Furthermore, the feature list of the behavioral classifiers contains features that are very setup specific (e.g., distance to arena wall, absolute coordinates etc.), or organism-line specific (e.g. mouse size/area). If the authors indeed aim at creating a standardized behavior classification pipeline, which can be compared across labs, it is essential to show/discuss how the method gives consistent results across at least some of the different experimental variants of this assay.

We agree with the reviewers that this is an important point, and we must stress that no model, including MARS, can be expected to work out of the box in all cases. We have found MARS to be stable across videos taken in different Caltech animal facilities, in different copies of our standardized behavior recording box, across an approximately five-year span of experiments; we expect comparable performance for any lab that records video in a setup akin to that of our previously published hardware.

When labs do not use this setup, we find that MARS sometimes does not produce satisfying results out of the box. We have therefore expanded our paper to include MARS_Developer, a separate python library for collecting of new pose annotation training data, and for fine-tuning the MARS detection and pose models to novel settings. We have tested this code on videos from several collaborators, and informally have found that MARS’s pose models can be finetuned to work in other top-view videos with varying arena sizes, backgrounds, and resolutions. For the purpose of the paper, we have included a demonstration of fine-tuning MARS to the previously published CRIM13 dataset (lines 364-369).

6) Although the paper describes the algorithm well, if was hard for the reviewers to judge the actual usability and quality of the tool without access to it. The software should be made available to the reviewers (with documentation)

We apologize for this omission. MARS is now publicly available at github.com/neuroethology/MARS, accompanied by installation instructions, download links for the trained MARS models, sample videos to test performance, and demo scripts. Several research groups have independently confirmed their ability to install MARS and run it on their data- if the reviewers encounter any difficulties, we are happy to assist in troubleshooting via the reviewing editor to ensure reviewer anonymity.

Similarly, documented code for training new behavior classifiers and fine-tuning the MARS detection and pose models is publicly available at github.com/neuroethology/MARS_Developer.

7) While a direct comparison to other social tracking methods (e.g., maDLC, SLEAP, others) is not necessary here, it is important to have a more comprehensive error analysis of the tracking (while the animals are in the bounding box, minimally).

We thank the reviewers for prompting this more thorough examination of the MARS pose estimation performance. We used the approach proposed by Ronchi and Perona (2017) to perform a detailed error analysis of MARS pose estimates; we have added mean Average Precision (mAP) and mean Average Recall (mAR) metrics to the text (lines 274-279, Table 1, Methods lines 286-326), as well as a new ED figure (ED Figure 4) in which we investigate pose errors in greater depth.

Given the large size of the MARS pose annotation dataset (15,000 frames, 5 annotators/frame), we were able to extract reasonable estimates of human annotation accuracy from the data, which we used to estimate the value of σ for the Object Keypoint Similarity metric (see text). We also computed mAP and mAR using a σ of 0.025, allowing direct comparison with the values reported in the SLEAP preprint. Lastly, we have created a Github repository MARS_pycocotools (https://github.com/neuroethology/MARS_pycocotools), which is a fork of the widely used COCO API for evaluation of detection and keypoint methods. Our fork allows users to evaluate their own pose models using the mouse pose sigmas estimated from MARS. We hope that this repository will encourage standardization of performance reporting in the animal pose estimation community, just as the COCO API has become a standard for the computer vision community.

8) Please ensure full statistical reporting in the main manuscript (e.g., t, F values, degrees of freedom, p value, etc.).

We thank the reviewers for catching this omission; we have included full statistical reporting in Table 2, and referenced this appropriately from the text.